# Modular Modeling of a Half-Vehicle System Using Generalized Receptance Coupling and Frequency-Based Substructuring (GRCFBS)

**Behzad Hamedi** [1,*] and **Saied Taheri** [2]

1    Engineering Mechanics Program, Mechanical Engineering Department, Virginia Polytechnic Institute and State University, Blacksburg, VA 24061, USA

2    Center for Tire Research (CenTiRe), Mechanical Engineering Department, Virginia Polytechnic Institute and State University, Blacksburg, VA 24061, USA; staheri@vt.edu

\*    Correspondence: behzadh@vt.edu

**Abstract:** This paper presents an advanced modular modeling approach for vertical vibration analysis of dynamic systems using the Generalized Receptance Coupling and Frequency-Based Substructuring (GRCFBS) method. The focus is on a four-DoF half-vehicle model comprising three key subsystems: front suspension, rear suspension, and the vehicle's trimmed body. The proposed technique is designed to predict dynamic responses in reconfigurable systems across various applications, including automotive, robotics, mechanical machinery, and aerospace structures. By coupling the receptance matrices (FRFs) of individual vehicle modules, the overall system receptance matrix is efficiently derived in a disassembled configuration. Two generalized coupling methods, originally developed by Jetmundsen and D.D. Klerk, are employed to determine the complete vehicle's receptance matrix from its subsystems. Validation is achieved by comparing the results with established methods, such as direct solution and modal analysis, demonstrating high accuracy and reliability for complex dynamic systems. This modular approach allows for the creation of reduced-order models focused on key measurement points without the need for detailed system representation. The method offers significant advantages in early-stage vehicle development, providing critical insights into system vibration behavior.

**Keywords:** Generalized Receptance Coupling (GRC); Frequency-Based Substructuring (FBS); half-vehicle model; modal analysis; dynamic response; receptance matrix; substructuring methods; reduced-order modeling; vibration analysis; reconfigurable systems



## 1. Introduction

Accurately predicting the dynamic behavior of engineering systems, particularly in the early development stages, remains a persistent challenge. Both numerical and experimental methods have inherent limitations that impact their effectiveness in modeling real-world systems. Numerical models offer flexibility but rely heavily on precise input data—such as geometry, material properties, boundary conditions, and contact characteristics—that are often difficult to obtain due to uncertainties. Conversely, experimental models provide a more realistic perspective but face constraints in spatial resolution, particularly at critical connection points, due to limited measurement capabilities. In the early stages of system development, the absence of physical prototypes exacerbates these difficulties, making both experimental validation and reliable numerical simulation challenging. Traditional approaches often fall short in predicting dynamic performance without sufficient input data or direct measurements. This highlights the need for a more modular, adaptable methodology capable of addressing incomplete data while delivering reliable predictions of system behavior. Frequency-Based Substructuring (FBS) has emerged as a powerful solution by breaking down complex systems into smaller, manageable subsystems, each

evaluated through receptance functions. This method allows for predicting the overall system's dynamic response by coupling the receptances of individual subsystems. FBS is particularly beneficial for systems with multiple interchangeable modules, where a limited number of reference points at subsystem interfaces serve as key excitation and measurement locations. Additionally, strategically selected internal measurement points within subsystems enhance the method's analytical flexibility. A notable advantage of FBS is its hybrid modeling capability, which combines numerical, experimental, and analytical data. This hybrid approach facilitates the integration of subsystems from diverse sources, ensuring accurate system-level predictions, even when data are incomplete or limited. For instance, FBS enables system-level vibration analysis with minimal subsystem data, even in the absence of detailed input data.

Integrating Generalized Receptance Coupling (GRC) with FBS extends dynamic modeling to reconfigurable systems across various industries, including automotive, aerospace, and robotics. In this study, the Generalized Receptance Coupling Frequency-Based Substructuring (GRCFBS) method is applied to a half-vehicle system with four degrees of freedom (DoF). This innovative approach focuses on the dynamic coupling of subsystems, such as the front suspension, rear suspension, and trimmed body. By coupling the individual subsystem receptances, the receptance matrix of the entire system is derived, enabling accurate dynamic predictions early in the design process without extensive physical prototyping or detailed input data. The effectiveness of this modular hybrid modeling approach is validated by comparing it with well-established numerical methods, including direct solution techniques and modal analysis. This study incorporates both Jetmundsen's and De Klerk's Lagrange Multiplier Frequency-Based Substructuring (LM-FBS) approaches, demonstrating how FBS can generate reduced-order models that maintain accuracy in critical areas while capturing the overall dynamic behavior of the system.

Historically, the groundwork for dynamic system analysis using impedance functions was laid by Bishop and Johnson (1960) [1] and later expanded by O'Hara (1961) [2], who applied these techniques to complex mechanical structures. Ewins and Gleeson (1975) [3] made significant contributions by deriving system parameters through Frequency Response Functions (FRFs), advancing the evolution of FBS. Building on these foundational works, Jetmundsen et al. (1980) [4] introduced a canonical form determining receptance of a system in terms of subsystems' receptance using FBS. In 2006, De Klerk et al. [5] introduced the Lagrange Multiplier Frequency-Based Substructuring (LM-FBS) method, which provides another canonical form allowing for indirect omission of certain FRFs at interface DoFs, minimizing the influence of noisy data. Additionally, Lagrange multipliers, which represent internal forces at connection points, can be obtained. This feature enhances LM-FBS's value in applications involving experimental FRF and requiring a reduction in noise uncertainty at substructure interfaces.

Zhang et al. (2017) [6] explored the dynamic interactions between vehicle bodies and subframes using FBS. His study explored the application of FBS to analyze the dynamic interactions between vehicle bodies and subframes. While the authors present valuable insights, they acknowledge the limitations in accurately modeling full-vehicle configurations due to the complexity of interactions and material properties. The work indicates a need for more comprehensive models that incorporate various dynamic effects and subsystems, which are often oversimplified in traditional approaches. However, Kang et al. (2019) [7] developed techniques for quantifying improvements in road noise through inverse substructuring. The paper discusses techniques for quantifying road noise improvements using inverse substructuring methods. However, the authors recognize that current methodologies often rely on simplified vehicle models that do not reflect real-world complexities. There is a clear need for more sophisticated full-vehicle models that can capture the intricacies of real driving conditions and their impact on NVH. Hülsmann et al. (2020) [8] applied dynamic substructuring to electric vehicles (EVs), addressing NVH issues specific to electric drivetrains. In their research, the authors applied dynamic substructuring techniques specifically for electric vehicles, focusing on NVH

issues related to electric drivetrains. The study highlights that, while advancements have been made, many existing models fail to account for the unique dynamic characteristics of electric vehicles, such as those arising from their powertrain and weight distribution. There is a gap in methodologies that adequately represent these dynamics in full-vehicle models. Tsai (2019) [9] advanced methods for measuring rotational receptance, crucial for rotational dynamics modeling. These developments reflect the growing utility of FBS, particularly for applications demanding more precise dynamic modeling. Additional contributions include Clontz and Taheri (2017) [10], who decoupled tire and suspension subsystems using FBS. Their research explored the decoupling of tire and suspension subsystems using FBS. While it provides insights into individual subsystem dynamics, it lacks a comprehensive approach to integrating these subsystems into a full-vehicle model for NVH analysis. The authors suggest that future work should focus on developing full-vehicle models that consider the interactions between various subsystems more holistically. Voormeeren and Rixen (2022) [11] examined the impact of measurement uncertainties on FBS outcomes. De Klerk, Rixen, and Jong (2021) [12] refined FBS with new algorithms for enhanced robustness, while Liu and Mir (2003) [13] explored hybrid approaches to vehicle axle noise prediction. FBS has proven its versatility across industries. Li et al. (2021) [14] applied FBS to railway vehicle dynamics, and Scheel and Sturzenegger (2020) [15] used it for satellite structural analysis in aerospace. In robotics, Hou et al. (2022) [16] employed FBS for modular robotic configurations, while Gebhardt et al. (2020) [17] applied it to wind turbine blade dynamics. Further, Park et al. (2023) [18] explored FBS in ship hull vibration analysis, and Lee et al. (2021) [19] demonstrated its relevance in civil engineering by analyzing large structures like bridges.

Since the foundational contributions of Jetmundsen and D.D. Klerk [20], FBS and receptance coupling methods have evolved considerably. Recent advancements include Schmitz et al. (2022) [21], who introduced an advanced approach merging Receptance Coupling Substructure Analysis (RCSA) with Bayesian machine learning for predicting milling stability, and Smith et al. (2021) [22], who developed an improved receptance matrix formulation that enhances precision for high-frequency dynamics and complex boundary conditions. Hou et al. (2023) [23] developed a framework for predicting FRFs in parameter-varying mechanical systems using generalized receptance coupling substructure analysis. In their paper, the authors develop a Multi-Body Dynamics (MBD) model for a full vehicle to analyze NVH. They note that, while their model improves upon traditional methods, challenges remain in integrating nonlinear characteristics of subsystems and ensuring accurate boundary conditions. This highlights the necessity for enhanced modeling techniques that can account for the complexities of full-vehicle dynamics.

Ji et al. (2018) [24] introduced a refined RCSA method for predicting tool tip dynamics, and De Klerk et al. (2021) [25] advanced FBS with algorithms that improve accuracy and efficiency in subsystem coupling. Hamedi and Taheri [26] reviewed hybrid modeling and modular substructuring using RCFBS, illustrating its effectiveness for vehicle noise and vibration prediction. Additionally, Hamedi and Taheri (2024) [27] provided a comparison of conventional modal analysis method suffering from mode truncation with the proposed RCFBS method, especially when dealing with high-frequency dynamics. In contrast, the RCFBS method provides greater accuracy when compared with numerical FEA and the direct method because it captures all relevant modes by working directly in the frequency domain, avoiding the truncation errors associated with modal analysis. This feature is particularly useful in systems with flexible or distributed parameters.

Despite these advancements, the development of modular FBS methods for vehicle dynamic modeling to study NVH performance, specifically for full-vehicle and half-vehicle car models, remains incomplete. This study introduces a novel, modular, FBS-based vibrational model for a half-vehicle, offering valuable insights during early development stages. This approach is particularly effective for target setting and cascading, providing a flexible, modular framework that emphasizes subsystem interactions and load paths for vibration transfer, especially in scenarios where numerical FEA models and experimental tests are

unavailable. This work demonstrates how GRCFBS can be employed to build efficient reduced-order models to handle the complexities of full-vehicle systems, particularly the interaction between subsystems like tire and suspensions.

This study introduces three key contributions:

1.  Advancing modular applications in vehicle models: Building on recent FBS applications, we enhance the modular capabilities of vehicle modeling by integrating both translational and rotational degrees of freedom (DoFs) for the vehicle body, enabling a more comprehensive representation of dynamic responses. In contrast to earlier studies, which primarily focused on translational DoFs, this dual DoF approach provides a holistic model of dynamic vehicle responses.

2.  Improved computational efficiency through sparse matrix representation: Consolidating substructures, like the front and rear suspensions, into a single module with sparse matrix representation reduces computational complexity and model sensitivity to noise, enhancing model reliability. This method minimizes off-diagonal term inversions, reducing unnecessary computational load and preserving boundary conditions.

3.  Robust validation of receptance data for predictive accuracy: By employing receptance-based FBS methods with both Jetmundsen's and LM-FBS algorithms, this study validates the capability of reduced-order models to capture resonant and anti-resonant behaviors critical for NVH studies.

    While both algorithms yield identical results when substructures are analytically determined, discrepancies may arise under experimental conditions. In Jetmundsen's formula, the receptance or FRFs at the interface directly influence subsystem coupling. The risk of noise propagation during experimental measurements arises because interface FRFs are susceptible to external influences, leading to potential inaccuracies. Conversely, the LM-FBS method mitigates these risks by managing the influence of interface FRFs through the product of the full receptance matrix and the Boolean matrix, thereby minimizing dependence on noisy data. Thus, for vehicle modeling, the LM-FBS method is superior when substructure receptance is derived from experimental measurements. This aspect is excluded from this study.

In summary, the GRCFBS approach demonstrates robust predictive accuracy for vehicle vibration modeling by introducing reliable reduced-order models that are computationally efficient for dynamic system identification in the early stages of development.

## 2. Technical Framework

### 2.1. Overview of Receptance Coupling Using Frequency-Based Substructuring (FBS)

Accurately determining the receptance of an assembly system is critical for predicting dynamic behavior in complex structural interactions. This framework provides a systematic approach to ensure the precise characterization and effective coupling of subsystems [27]. The process begins by setting the dynamic analysis objectives and identifying key points of interest, such as critical excitation and measurement points, which help define the generalized coordinates and degrees of freedom (DoFs) at significant nodes. Following this, the structure is segmented into distinct subsystems based on their dynamic properties, including natural frequencies and boundary conditions. This segmentation allows for an independent analysis of each subsystem prior to coupling.

The next step involves accurately modeling the connections between these subsystems. Whether these connections are simple single-point, multi-point, or complex, such as bushings, it is essential to account for their stiffness, damping, and kinematic relationships. Once the connections are modeled, generalized coordinates and DoFs are allocated at both the connection points and the internal points within each subsystem, focusing on modes of motion that are essential for capturing the dynamic behavior. Receptance matrices for the subsystems are then determined using a combination of experimental data, numerical simulations such as Finite Element Analysis (FEA), or analytical methods tailored to the complexity of the subsystem.

To improve the accuracy of the experimental receptance data, filtering techniques are applied to the Frequency Response Function (FRF) data to remove noise. The next step is to select an appropriate coupling method based on the specific requirements of the analysis. This could involve direct receptance coupling, modal-based coupling, or more advanced methods, such as Lagrange Multiplier Frequency-Based Substructuring (LM-FBS) or the Jetmundsen method. Once the coupling method is selected, the direct and cross components of the receptance matrix are calculated, ensuring equilibrium and continuity across the subsystems. Finally, these components are combined to construct the complete assembly system's receptance matrix, which encapsulates the dynamic characteristics of the entire structure and predicts its response to external excitations.

The Receptance Coupling Frequency-Based Substructuring (RCFBS) method offers several advantages for dynamic analysis, particularly when compared to traditional methods such as the modal method. FBS-focused methods are preferable to conventional methods in scenarios where the systems are modular or reconfigurable and in cases where detailed FEM or MBD models are unavailable or impractical. This method also allows for combining FRF data from various sources, such as physical tests, numerical simulations, and analytical models, which is not feasible with conventional approaches. RCFBS enables reduced-order modeling by focusing on critical points of interest—namely connection points and essential internal nodes—thereby reducing computational complexity while maintaining high accuracy in these significant areas. In addition, unlike traditional modal methods that are based on mode truncation, RCFBS captures the full dynamic behavior of substructures by utilizing full receptance matrices, particularly improving accuracy in higher-frequency ranges with subsystems with distributed parameters.

Another key advantage of RCFBS is its flexibility, as the method allows for deriving receptance matrices from both experimental and numerical data, such as FEA. This versatility makes it suitable for handling complex interactions or boundary conditions. This also provides an advantage when physical or numerical measurements at connection points are inaccessible, as the method can still predict the dynamic response at these points. Additionally, the direct use of receptance data ensures accurate representation of dynamic interactions, especially at higher frequencies where subsystem interactions are more complex. RCFBS also integrates advanced coupling techniques, such as those developed by Jetmundsen and D.D. Klerk, further enhancing its applicability for dynamic modeling in systems with multiple substructures and degrees of freedom.

### 2.2. Half-Vehicle Model

The half-car model serves as a classic example in dynamic analysis, especially for evaluating the effects of suspension systems on ride performance. The suspension subsystem plays a critical role in ensuring vehicle comfort and stability. Figure 1 illustrates the schematic of the half-car model, which includes a trimmed body with mass $m_s$ and moment of inertia $I_{CG}$. The body interfaces with the front and rear suspensions at designated mounting points, represented by generalized coordinates $u_1$ and $u_2$. These coordinates are used in the model instead of pitch angle and bounce degrees of freedom (DoFs) of the center of gravity, as they better reflect the connection points.

The front and rear suspensions are modeled with linear springs and dampers, characterized by stiffness coefficients $K_{sf}$ and $K_{sr}$ and damping coefficients $C_{sf}$ and $C_{sr}$, respectively. Unsprung masses are represented by lumped masses $m_{uf}$ and $m_{ur}$ at the wheel centers, with generalized coordinates $u_3$ and $u_4$. The wheel–ground contact is modeled using tire characteristics with stiffness $K_{tf}$ and $K_{tr}$ and damping coefficients $C_{tf}$ and $C_{tr}$.

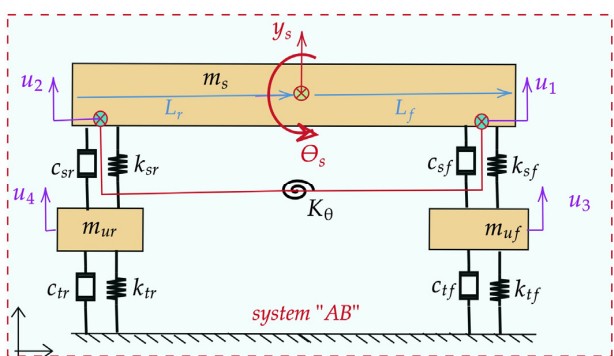

**Figure 1.** Half-car model.

The angular and vertical displacements of the trimmed body are depicted in Figure 1, with vertical coordinates representing displacements at both ends of the vehicle. Analytical development of the mass, damping, and stiffness matrices for this model is possible. Consequently, the system features four generalized coordinates corresponding to vertical displacement DoFs. The vehicle is positioned on a flat surface and excited at key points to measure displacements at connection and internal points, like modal experimental measurements.

Like using an anti-roll bar to model roll stiffness, many vehicle models incorporate an anti-pitch bar or equivalent torsional spring mechanism to account for pitch stiffness [27]. This approach, as supported in various vehicle dynamics texts [Jazar and Marzbani, 2024], allows for the model to capture pitch resistance by adding a torsional stiffness element, simulating the vehicle's response to dynamic forces without introducing additional dynamics due to the mass of a physical component. The torsional stiffness (kθ) of the anti-pitch bar defines resistance against pitch rotation. Pitch stiffness in a vehicle depends on various factors, including suspension geometry, underbody stiffness, subframe rigidity, and components like anti-pitch bars or torsional mechanisms.

The challenge lies in determining whether subsystems, such as the front and rear suspensions along with the trimmed body, can be measured in free-free conditions to obtain their Frequency Response Functions (FRFs). This approach simplifies access to measurement points and allows for the coupling of subsystem responses and FRFs to derive the overall car receptance matrix.

While the half-vehicle model can be solved using direct methods (exact solution), modal methods, and determinant methods, this paper focuses on solving it through receptance coupling with Frequency-Based Substructuring (FBS). The first step involves decomposing the system into substructures. In this case, the vehicle is divided into three subsystems: the front suspension, rear suspension, and trimmed body, as shown in Figure 2. To enhance model reliability, the top mounts of the suspensions are replaced by a dummy mass $m_d$ connected to the upper body with a rigid link. Consequently, the vehicle system dynamics can be resolved by coupling the dynamics of the three subsystems: A, $B_1$, and $B_2$.

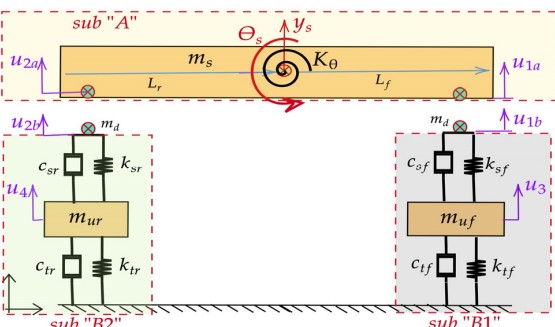

**Figure 2.** Initial substructuring scheme of a car system to three substructures.

To streamline the coupling process, a reduced-order substructuring approach is employed, merging the front and rear suspensions into a single subsystem B. This reduces the problem from coupling three subsystems to coupling two subsystems, simplifying the mathematical model and formulation. Figure 3 illustrates this condensed substructuring scheme, showing the trimmed body as substructure A and the combined suspension subsystem as substructure B. Generalized coordinates and forces at the connection points are also depicted.

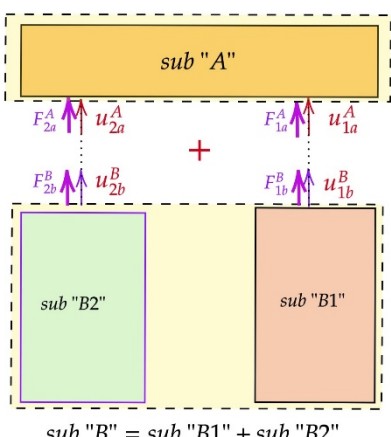

**Figure 3.** Reduced-order substructuring scheme to two substructures.

### 2.3. FBS-Based Generalized Receptance Coupling (GRC) Technique

The Generalized Receptance Coupling (GRC) method is an advanced extension of traditional receptance coupling techniques, designed to handle systems composed of multiple interconnected substructures. Rooted in Frequency-Based Substructuring (FBS), this method enables the accurate prediction of system receptances in complex assemblies of N subsystems. In receptance coupling, the boundary conditions of displacement compatibility and force equilibrium at connection interfaces are crucial. While rigid connections are often assumed for simplicity, interface flexibility can also be incorporated. This section outlines two prominent GRC methods: the Lagrange Multiplier Frequency-Based Substructuring (LM-FBS) method and the Jetmundsen algorithm. Although a basic two-substructure coupling example is used to illustrate these methods, both approaches can be extended to more complex assemblies.

### 2.3.1. Generalized Coupling Method Using LM-FBS

The LM-FBS method employs Lagrange multipliers to introduce internal forces at connection points between substructures, ensuring compatibility and force equilibrium at these interfaces. The core formulation of the method revolves around the computation of the coupled receptance matrix, which is obtained by incorporating the dynamics of individual substructures along with any interface compliance. The general formula for the coupled receptance matrix is given by [5,12]:

$$[H]^{AB} = [H] - [H][B]^{T}\left([B][H][B]^{T} + [H]_{bush}\right)^{-1}[B][H] \tag{1}$$

where [B] is the Boolean matrix that ensures compatibility of the displacements at the connection interfaces. $[H]^{AB}$ is the receptance matrix of the assembled system, and [H] contains the uncoupled receptance matrix of the substructures. $[H_{bush}]$ accounts for the compliance of the interface (if applicable), which can be modeled as follows:

$$[H_{bush}] = \left([K] + j\omega[C] - \omega^2[M]\right)_{bush}^{-1} \tag{2}$$

This formulation accounts for interface compliance, which is modeled as a form of compatibility relaxation, accurately capturing the dynamic behavior of interconnected subsystems. The interface forces, λ, that balance the substructures are given by the following:

$$\{\lambda\} = \left([B][H][B]^{\mathrm{T}} + [H_{bush}]\right)^{-1}[B][H]\{F\} \qquad (3)$$

For a system with two substructures—substructure A (trimmed body) with two DoFs and substructure B (platform with front and rear suspensions) with four DoFs—the $6 \times 6$ uncoupled receptance matrix is structured as follows:

$$[H]_{6\times6} = \begin{bmatrix} \left[H^{A}_{\overline{AA}}\right]_{2\times2} & 0 \\ 0 & \left[H^{B}_{\overline{BB}}\right]_{4\times4} \end{bmatrix} \qquad (4)$$

where $\left[H^{A}_{\overline{AA}}\right]$ is the internal receptance matrix of substructure A, with both internal and connection DoFs. $\left[H^{B}_{\overline{BB}}\right]$ represents the internal receptance matrix of substructure B. The receptance matrix of substructures A and B is strctured as follows:

$$\left[H^{A}_{\overline{AA}}\right] = \begin{bmatrix} H^{A}_{11} & H^{A}_{12} \\ H^{A}_{21} & H^{A}_{22} \end{bmatrix},$$

$$\left[H^{B}_{\overline{BB}}\right] == \begin{bmatrix} H^{B}_{33} & H^{B}_{34} & H^{B}_{31b} & H^{B}_{32b} \\ H^{B}_{43} & H^{B}_{44} & H^{B}_{41b} & H^{B}_{42b} \\ H^{B}_{1b3} & H^{B}_{1b4} & H^{B}_{1b1b} & H^{B}_{1b2b} \\ H^{B}_{2b3} & H^{B}_{2b4} & H^{B}_{2b1} & H^{B}_{2b2b} \end{bmatrix} = \begin{bmatrix} H^{B}_{33} & 0 & H^{B}_{31b} & 0 \\ 0 & H^{B}_{44} & 0 & H^{B}_{42b} \\ H^{B}_{1b3} & 0 & H^{B}_{1b1b} & 0 \\ 0 & H^{B}_{2b4} & 0 & H^{B}_{2b2b} \end{bmatrix} \qquad (5)$$

Since there is no internal coupling between the front and rear suspension subsystems, several off-diagonal elements are zero:

$$H^{B}_{34} = H^{B}_{43} = H^{B}_{32b} = H^{B}_{2b3} = H^{B}_{41b} = H^{B}_{1b4} = H^{B}_{1b2b} = H^{B}_{2b1b} = 0 \qquad (6)$$

The Boolean matrix [B], which ensures displacement compatibility, is constructed as follows:

$$\underbrace{\begin{bmatrix} 1 & 0 & 0 & 0 & -1 & 0 \\ 0 & 1 & 0 & 0 & 0 & -1 \end{bmatrix}}_{[B]} \begin{Bmatrix} u^{A}_{1a} \\ u^{A}_{2a} \\ u^{B}_{3} \\ u^{B}_{4} \\ u^{B}_{1b} \\ u^{B}_{2b} \end{Bmatrix} = [0] \qquad (7)$$

This matrix enforces the conditions $u^{A}_{1a} = u^{B}_{1b}$ and $u^{A}_{2a} = u^{B}_{2b}$, ensuring displacement compatibility at the connection points. Substituting [H] and [B] into the equations of motion, the resulting coupled receptance matrix $[H]^{AB}$ describes the dynamic response of the system. The displacement–force relationship for the assembled system can be expressed as follows:

$$\begin{Bmatrix} u_{1a} \\ u_{2a} \\ u_{3} \\ u_{4} \\ u_{1b} \\ u_{2b} \end{Bmatrix}^{AB} = [H]^{AB} \begin{Bmatrix} F_{1a} \\ F_{2a} \\ F_{3} \\ F_{4} \\ F_{1b} \\ F_{2b} \end{Bmatrix}^{AB} \qquad (8)$$

where $u^{A}_{1a}$ and $u^{A}_{2a}$ are the displacements at the connection points of substructure A and, similarly, $u^{B}_{1b}$ and $u^{B}_{2b}$ are the corresponding displacements at the connection points of

substructure B. $F_{1a}$, $F_{2a}$, $F_{1b}$, and $F_{2b}$ are the forces applied at these connection points, and $F_3$ and $F_4$ are the internal forces within substructure B.

In this system of equations, two of the six equations are redundant, as they describe identical conditions for the displacements at the connection points, ensuring continuity between substructures A and B. Specifically, the compatibility equations ensure that the displacements are continuous across the interface, maintaining the physical condition of the coupled system.

Thus, the reduced system reflects the overall dynamic behavior of the interconnected subsystems, with the coupled receptance matrix $[H]^{AB}$ encapsulating the effects of both internal dynamics and interface coupling.

### 2.3.2. Jetmundsen Algorithm

The Jetmundsen algorithm computes the coupled receptance matrix by utilizing the receptance matrices of subsystems, considering both internal and connection degrees of freedom (DoFs) and involving the total receptance at the connection points. For a system composed of two substructures, A and B, the equations of motion for the decoupled substructures are given as follows:

$$\{u^A\} = [H^A]\{F^A\}, \quad \{u^B\} = [H^B]\{F^B\} \tag{9}$$

where $[H^A]$ and $[H^B]$ are the receptance matrices of subsystems A and B, respectively. When coupling substructures A and B, Jetmundsen's general algorithm simplifies to the following form [4]:

$$[H]^{AB} = \begin{bmatrix} [H_{\overline{AA}}] & [0] \\ [0] & [H_{BB}] \end{bmatrix} - \begin{bmatrix} [H_{\overline{A}c}] \\ -[H_{Bc}] \end{bmatrix} \left[[H_{cc}^A] + [H_{cc}^B]\right]^{-1} \begin{bmatrix} [H_{\overline{A}c}] \\ -[H_{Bc}] \end{bmatrix}^T \tag{10}$$

In this expression,

- $\overline{AA}$ indicates that substructure A retains both internal and interface DoFs, whereas substructure B contains only internal DoFs. This differs from earlier coupling methods where both substructures retained generalized coordinates at the interface.
- $[H_{cc}^A]$ and $[H_{cc}^B]$ represent the receptance matrices at the connection points between the two substructures.

The term $\left[[H_{cc}^A] + [H_{cc}^B]\right]^{-1}$ dominates the computational cost but remains relatively small in size, making the Jetmundsen algorithm computationally efficient. This matrix represents the total receptance at the connection points, assuming a rigid interface. As the size of this matrix is significantly smaller than the subsystem receptance matrices, the computational load is reduced. If interface flexibility is present, such as a bushing, its receptance can be incorporated as $\left[H_{cc}^A + H_{cc}^B + [H_{bush}]\right]^{-1}$, further refining the coupled system's dynamic response. In a simplified case with a rigid interface, the full coupled receptance matrix is computed as follows:

$$[H]^{AB} = \begin{bmatrix} H_{11}^A & H_{12}^A & 0 & 0 \\ H_{21}^A & H_{22}^A & 0 & 0 \\ 0 & 0 & H_{33}^B & 0 \\ 0 & 0 & 0 & H_{44}^B \end{bmatrix}$$
$$- \begin{bmatrix} H_{11}^A & H_{12}^A \\ H_{21}^A & H_{22}^A \\ -H_{31}^B & 0 \\ 0 & -H_{42}^B \end{bmatrix} \left[\begin{bmatrix} H_{11}^A & H_{12}^A \\ H_{21}^A & H_{22}^A \end{bmatrix} + \begin{bmatrix} H_{11}^B & H_{12}^B \\ H_{21}^B & H_{22}^B \end{bmatrix}\right]^{-1} \begin{bmatrix} H_{11}^A & H_{12}^A \\ H_{21}^A & H_{22}^A \\ -H_{31}^B & 0 \\ 0 & -H_{42}^B \end{bmatrix}^T \tag{11}$$

By calculating the coupled receptance matrix $[H]^{AB}$, the dynamic response of the assembled system can be determined. The relationship between the displacements and applied forces at the generalized coordinates is given by the following:

$$\begin{Bmatrix} u_1 \\ u_2 \\ u_3 \\ u_4 \end{Bmatrix}^{AB} = \begin{bmatrix} H_{11}^{AB} & H_{12}^{AB} & H_{13}^{AB} & H_{14}^{AB} \\ H_{21}^{AB} & H_{22}^{AB} & H_{23}^{AB} & H_{24}^{AB} \\ H_{31}^{AB} & H_{32}^{AB} & H_{33}^{AB} & H_{34}^{AB} \\ H_{41}^{AB} & H_{42}^{AB} & H_{43}^{AB} & H_{44}^{AB} \end{bmatrix} \begin{Bmatrix} F_1 \\ F_2 \\ F_3 \\ F_4 \end{Bmatrix}^{AB} \tag{12}$$

Here, $[H]^{AB}$ represents the full receptance matrix of the assembled system. The subscripts A and B correspond to the upper substructure (trimmed body) and the lower substructure (platform consisting of front and rear unsprung mass elements), respectively. These receptance matrices are typically measured under free-free boundary conditions to ensure accurate subsystem characterization. By leveraging this approach, the Jetmundsen algorithm allows for an efficient and accurate prediction of the dynamic response of coupled systems, such as vehicle subsystems.

### 2.4. Substructuring and Determining Receptance Functions for Subsystems

In dynamic system analysis, especially for complex multi-body systems such as vehicles, the behavior of individual subsystems contributes significantly to the overall dynamic response. Substructuring techniques allow for the isolation of these subsystems, enabling a detailed analysis of each component's dynamic characteristics. By calculating the receptance functions—also referred to as frequency response functions (FRFs)—of each subsystem, we can assess their individual dynamic behaviors and, later, couple them to predict the entire system's response. This approach is particularly advantageous in reducing the computational complexity of the system while maintaining accuracy, as the subsystems can be modeled using various methods, including analytical derivation, numerical techniques like Finite Element Analysis (FEA), or experimental measurements.

The equations outlined in Equations (1) and (10) present the mathematical framework for determining the coupled system receptances by utilizing the receptance matrices of individual subsystems. In this study, a four DoF half-car model is employed, where the trimmed body (representing the sprung mass) and the front and rear unsprung masses are treated as two degrees of freedom (DoF) systems. This allows for the analytical derivation of their respective receptance components. However, for more complex subsystems, receptance components are typically determined through numerical methods, such as Finite Element Analysis (FEA) or Multi-Body Dynamics (MBD), or experimentally using Frequency Response Function (FRF) measurements.

#### 2.4.1. Front Suspension Receptance Derivation

In this section, we focus on deriving the receptance functions for the front suspension subsystem, which plays a critical role in defining the vehicle's dynamic response. The suspension connects the vehicle body to the unsprung mass and dictates how external forces—such as road disturbances—are transmitted to the body. By deriving the receptance functions, we can evaluate the front suspension's behavior under dynamic loading and its contribution to the overall system response. For the front suspension subsystem, the equations of motion are expressed as follows:

$$\begin{cases} m_d \ddot{u}_{1b}^B + C_{sf} \left( \dot{u}_{1b}^B - \dot{u}_3^B \right) + K_{sf} \left( u_{1b}^B - u_3^B \right) = F_{1b}^B \\ m_{uf} \ddot{u}_3^B + C_{sf} \left( \dot{u}_3^B - \dot{u}_{1b}^B \right) + K_{sf} \left( u_3^B - u_{1b}^B \right) = F_3^B \end{cases} \tag{13}$$

where $m_d$ is the dummy mass added at the suspension top, $m_{uf}$ is the unsprung mass of the front suspension, and $C_{sf}$ and $K_{sf}$ are the damping and stiffness of the suspension,

respectively. By applying the Laplace transform and assuming an excitation force applied at the connection point of the front suspension, the following receptance functions are derived:

$$H_{31}^B = \frac{U_3^B}{F_1^B}, \ H_{11}^B = \frac{U_1^B}{F_1^B} \tag{14}$$

Similarly, applying the excitation force $F_3^B$ and measuring displacements at the rear suspension coordinates $u_2^B$ and $u_4^B$, we obtain the following:

$$H_{33}^B = \frac{U_3^B}{F_3^B}, \ H_{13}^B = \frac{U_1^B}{F_3^B} \tag{15}$$

As expected from the reciprocity principle, it holds that $H_{31}^B = H_{13}^B$. The corresponding analytical expressions for these receptance components are the following:

$$\mathrm{H}_{11}^B = \frac{p_1}{p_1 p_2 - p_3^2}, \mathrm{H}_{13}^B = \mathrm{H}_{31}^B = \frac{p_3}{p_1 p_2 - p_3^2}, \mathrm{H}_{33}^B = \frac{p_2}{p_1 p_2 - p_3^2} \tag{16}$$

where the parameters $p_1$, $p_2$, and $p_3$ are defined as follows:

$$\begin{aligned}
p_1 &= -m_{uf}\omega^2 + \left(K_{sf} + K_{tf}\right) + \left(C_{sf} + C_{tf}\right)\omega j \\
p_2 &= -m_d\omega^2 + C_{sf}\omega j + K_{sf} \\
p_3 &= C_{sf}\omega j + K_{sf}
\end{aligned} \tag{17}$$

### 2.4.2. Rear Suspension Receptance Derivation

In this section, we extend the analysis to the rear suspension subsystem, which, along with the front suspension, governs the vehicle's ride dynamics and stability. The rear suspension (substructure $B_2$, part of substructure B) is responsible for managing the dynamic loads acting on the rear of the vehicle, particularly in response to road surface variations and vehicle acceleration or braking. Deriving the receptance functions for the rear suspension allows for us to understand its influence on the overall system's dynamic response, complementing the front suspension's role. For the rear suspension, the equations of motion are similarly expressed as follows:

$$\begin{cases}
m_d\ddot{u}_{2b}^B + C_{sr}\left(\dot{u}_2^B - \dot{u}_4^B\right) + K_{sf}\left(u_2^B - u_4^B\right) = F_2^B \\
m_{ur}\ddot{u}_4^B + C_{sr}\left(\dot{u}_4^B - \dot{u}_2^B\right) + K_{sr}\left(u_4^B - u_2^B\right) = F_4^B
\end{cases} \tag{18}$$

Here, $C_{sr}$ and $K_{sr}$ represent the damping and stiffness of the rear suspension, while $m_{ur}$ is the unsprung mass. The derived receptance components are the following:

$$H_{22}^B = \frac{q_1}{q_1 q_2 - q_3^2}, H_{24}^B = H_{42}^B = \frac{q_3}{q_1 q_2 - q_3^2}, H_{44}^B = \frac{q_2}{q_1 q_2 - q_3^2} \tag{19}$$

with the parameters $q_1$, $q_2$, $q_3$ defined as follows:

$$\begin{aligned}
q_1 &= -m_{ur}\omega^2 + (K_{sr} + K_{tr}) + (C_{sr} + C_{tr})\omega j \\
q_2 &= -m_d\omega^2 + C_{sr}\omega j + K_{sr} \\
q_3 &= C_{sr}\omega j + K_{sr}
\end{aligned} \tag{20}$$

Thus, Equations (5), (6), and (19) provide the receptance matrix components for substructure B, encompassing both front and rear suspension subsystems.

### 2.4.3. Trimmed Body (Substructure A) Receptance Derivation

Next, we turn to substructure A, representing the vehicle's trimmed body, often referred to as the sprung mass. The trimmed body plays a central role in determining

overall ride comfort and stability, as it directly interacts with both the front and rear suspension subsystems. A key aspect of this analysis is the body's rotational stiffness in the pitch direction, which governs its resistance to rotational motion about the center of gravity. This stiffness is crucial for accurately capturing the vehicle's pitch dynamics and modeling its response to vertical forces at the suspension interfaces.

In this analysis, both vertical (heave) and rotational (pitch) modes of motion are considered to provide a comprehensive understanding of how forces transmitted through the suspension influence the body structure's overall motion. The vertical (heave) motion describes the upward and downward movement of the vehicle body, while the rotational (pitch) motion involves rotation about the vehicle's lateral axis. To model these dynamics, Newton's second law of motion is applied to both the translational and rotational movements. In the vertical (heave) direction, the summation of forces acting on the vehicle body is equal to the mass times the acceleration of the center of gravity. Similarly, for rotational (pitch) motion, the summation of moments about the center of gravity is equal to the product of the body's mass moment of inertia and its angular acceleration. These governing principles can be mathematically expressed as follows:

$$\begin{cases} \sum F_y^A = m_s \ddot{y}_s \\ \sum M_{CG}^A = I_{CG} \ddot{\theta}_s \end{cases} \tag{21}$$

where $m_s$ is the mass of the sprung body, $I_{CG}$ is the mass moment of inertia at the center of gravity, and $\theta s$ is the angular displacement. The generalized coordinates $u_1^A$ and $u_2^A$ correspond to displacements at the front and rear suspension mounting points. The transformation to generalized coordinates is given by the following:

$$\theta_s = \frac{u_1^A - u_2^A}{L}, \ y_s = \frac{L_r}{L} u_1^A + \frac{L_f}{L} u_2^A \tag{22}$$

where $L_f$ and $L_r$ are the distances from the center of gravity to the front and rear suspensions, respectively. Assuming an excitation force $F_2^A$ applied at the rear connection point and combining Equations (21) and (22), the equations of motion become the following:

$$\begin{cases} F_2^A = m_s \left( \frac{L_r}{L} \ddot{u}_1^A + \frac{L_f}{L} \ddot{u}_2^A \right) \\ -F_2^A L_r - k_\theta \left( \frac{u_1^A - u_2^A}{L} \right) = I_{CG} \left( \frac{\ddot{u}_1^A - \ddot{u}_2^A}{L} \right) \end{cases} \tag{23}$$

Transforming the equations to the frequency domain and deriving $u_2^A$ and $u_1^A$ as functions of $F_2^A$, the displacement receptance components $H_{22}^A$, and $H_{12}^A$ are determined as follows:

$$H_{22}^A = \frac{u_2^A}{F_2^A} = -\left( \frac{1}{m_s \omega^2} - \frac{L_r^2}{k_\theta - I_{CG} \omega^2} \right)$$
$$H_{12}^A = \frac{u_1^A}{F_2^A} = -\left( \frac{1}{m_s \omega^2} + \frac{L_f L_r}{k_\theta - I_{CG} \omega^2} \right) \tag{24}$$

Following a similar approach for an excitation force $F_1^A$, we obtain the following:

$$H_{11}^A = \frac{u_1^A}{F_1^A} = -\left( \frac{1}{m_s \omega^2} - \frac{L_f^2}{k_\theta^A - I_{CG} \omega^2} \right)$$
$$H_{21}^A = \frac{u_2^A}{F_1^A} = -\left( \frac{1}{m_s \omega^2} + \frac{L_f L_r}{k_\theta - I_{CG} \omega^2} \right) \tag{25}$$

These equations satisfy the reciprocity principle, as $H_{21}^A = H_{12}^A$.

The torsional spring $k_\theta$ provides a simplified method for introducing pitch resistance without altering the rigid body dynamics of the vehicle. The upper body remains rigid, and the torsional spring merely captures the pitch stiffness without compromising the overall assumption of a rigid body. In this approach, pitch stiffness has been modeled with a torsional spring between the front and rear body attachments using two massless rigid

beams connected by a torsional spring to account for the effects of pitch stiffness similar to modeling roll stiffness, as demonstrated in Figure 1.

*2.5. Validation: Correlation Using Classical Methods*

Validation of dynamic system models is essential to ensure the accuracy and reliability of the predictions made by the model. This section describes the methods used for validating the model's receptance coupling matrix, employing classical techniques including dynamic system modeling, determinant-based methods, and modal analysis.

2.5.1. Dynamic System Modeling and Direct Solution Approach

In this subsection, the direct method (exact solution) for determining the receptance coupling matrix is examined. This approach involves formulating the system equations of motion using classical techniques and solving them in the Laplace domain. We specifically employ the Lagrange method, which is well-suited for complex dynamic systems with multiple degrees of freedom due to its energy-based formulation.

The Lagrange method is selected for its effectiveness in deriving equations of motion by utilizing the principles of energy. This method is advantageous when dealing with systems that have multiple interacting components and generalized coordinates. By expressing the system's kinetic and potential energies in terms of these coordinates, the equations of motion can be derived in a systematic way. For the vehicle system with generalized coordinates $u_1$, $u_2, u_3$, $u_4$, the kinetic energy $T$, damping $D$, and potential energy $V$ are defined as follows:

$$
\begin{aligned}
T &= \tfrac{1}{2}\left\{ m_s\left(\tfrac{L_r\dot{u}_1+L_f\dot{u}_2}{L}\right)^2 + I_{CG}\left(\tfrac{\dot{u}_1-\dot{u}_2}{L}\right)^2 + m_{uf}\dot{u_3}^2 + m_{ur}\dot{u_4}^2 \right\}\\
D &= \tfrac{1}{2}\left\{ C_{sf}\left(\dot{u}_1-\dot{u}_3\right)^2 + C_{sr}\left(\dot{u}_2-\dot{u}_4\right)^2 + C_{tf}\dot{u_3}^2 + C_{tr}\dot{u_4}^2 \right\}\\
V &= \tfrac{1}{2}\left\{ K_{sf}(u_1-u_3)^2 + K_{sr}(u_2-u_4)^2 + K_{tf}u_3{}^2 + K_{tr}u_4{}^2 + k_\theta\left(\tfrac{u_1-u_2}{L}\right)^2 \right\}
\end{aligned}
\tag{26}
$$

The equations of motion are derived from the Lagrange equations:

$$
\frac{dy}{dx}\left(\frac{\partial T}{\partial \dot{u}_i}\right) - \frac{\partial T}{\partial u_i} + \frac{\partial V}{\partial u_i} + \frac{\partial D}{\partial \dot{u}_i} = Q_i
\tag{27}
$$

where $u_i$ and $Q_i$ are generalized coordinates and generalized external force, respectively. Equations (26) and (27) lead to the following:

$$
\begin{cases}
\alpha_1\ddot{u}_1 + \alpha_2\ddot{u}_2 + C_{sf}\left(\dot{u}_1-\dot{u}_3\right) + K_{sf}(u_1-u_3) + \frac{k_\theta}{L^2}(u_1-u_2) = F_1\\
\alpha_2\ddot{u}_1 + \alpha_1\ddot{u}_2 + C_{sf}\left(\dot{u}_2-\dot{u}_4\right) + K_{sf}(u_2-u_4) - \frac{k_\theta}{L^2}(u_1-u_2) = F_2\\
m_{uf}\ddot{u}_3 + \left(K_{sf}+K_{tf}\right)u_3 + \left(C_{sf}+C_{tf}\right)\dot{u}_3 - K_{sf}u_1 - C_{sf}\dot{u}_1 = F_3\\
m_{ur}\ddot{u}_4 + (K_{sr}+K_{tr})u_4 + \left(C_{sf}+C_{tf}\right)\dot{u}_4 - K_{sr}u_2 - C_{sr}\dot{u}_2 = F_3
\end{cases}
\tag{28}
$$

The matrix form of the equations of motion is the following:

$$
\begin{bmatrix}
\alpha_1 & \alpha_2 & 0 & 0\\
\alpha_2 & \alpha_1 & 0 & 0\\
0 & 0 & m_{uf} & 0\\
0 & 0 & 0 & m_{ur}
\end{bmatrix}
\begin{Bmatrix}
\ddot{u}_1\\ \ddot{u}_2\\ \ddot{u}_3\\ \ddot{u}_4
\end{Bmatrix}
+
\begin{bmatrix}
C_{sf} & 0 & -C_{sf} & 0\\
0 & C_{sr} & 0 & -C_{sr}\\
-C_{sf} & 0 & C_{sf}+C_{tf} & 0\\
0 & -C_{sr} & 0 & C_{sr}+C_{tr}
\end{bmatrix}
\begin{Bmatrix}
\dot{u}_1\\ \dot{u}_2\\ \dot{u}_3\\ \dot{u}_4
\end{Bmatrix}
$$
$$
+
\begin{bmatrix}
K_{sf}+\frac{k_\theta}{L^2} & -\frac{k_\theta}{L^2} & -K_{sf} & 0\\
-\frac{k_\theta}{L^2} & K_{sr}+\frac{k_\theta}{L^2} & 0 & -K_{sr}\\
-K_{sf} & 0 & K_{sf}+K_{tf} & 0\\
0 & -K_{sr} & 0 & K_{sr}+K_{tr}
\end{bmatrix}
\begin{Bmatrix}
u_1\\ u_2\\ u_3\\ u_4
\end{Bmatrix}
=
\begin{Bmatrix}
F_1\\ F_2\\ F_3\\ F_4
\end{Bmatrix}
\tag{29}
$$

Assuming harmonic excitation forces, the steady-state response can be obtained. The linear equations of motion for the assembly system AB are expressed as follows:

$$\begin{Bmatrix} u_1^{AB} \\ u_2^{AB} \\ u_3^{AB} \\ u_4^{AB} \end{Bmatrix} = [H^{AB}] \begin{Bmatrix} F_1 \\ F_2 \\ F_3 \\ F_4 \end{Bmatrix} \tag{30}$$

The receptance matrix $[H^{AB}]$ of the assembly system AB is determined as follows:

$$[H^{AB}] = \begin{bmatrix} H_{11}^{AB} & H_{12}^{AB} & H_{13}^{AB} & H_{14}^{AB} \\ H_{21}^{AB} & H_{22}^{AB} & H_{23}^{AB} & H_{24}^{AB} \\ H_{31}^{AB} & H_{32}^{AB} & H_{33}^{AB} & H_{34}^{AB} \\ H_{41}^{AB} & H_{42}^{AB} & H_{43}^{AB} & H_{44}^{AB} \end{bmatrix} = \left[ -\omega^2[M_{AB}] + j\omega[C_{AB}] + [K_{AB}] \right]^{-1}. \tag{31}$$

where $[M_{AB}]$, $[C_{AB}]$, and $[K_{AB}]$ represent the mass, damping, and stiffness matrices of the assembly system, respectively. This receptance matrix captures the system's dynamic response characteristics and is crucial for analyzing and predicting the vibrational behavior of the coupled system.

### 2.5.2. Determinant-Based Method

Although this method is not a standalone approach, it serves as a computational technique within the broader framework of calculating the receptance matrix. The determinant-based method is a classical technique often employed in multi-degree-of-freedom systems to compute individual elements of the Frequency Response Function (FRF) matrix [1]. While it shares some similarities with matrix inversion techniques, this method leverages Cramer's Rule and Cofactor Expansion, which offer distinct advantages in certain applications. Specifically, the method calculates the FRF for a given degree of freedom as the ratio of two determinants: the determinant of a modified dynamic stiffness matrix (with the i-th row and j-th column removed) and the determinant of the full dynamic stiffness matrix. In contrast to a general matrix inversion approach, which computes all elements of the inverse matrix simultaneously, the determinant-based method focuses on individual matrix elements, offering a more targeted and computationally efficient solution, especially when FRFs for specific degrees of freedom are required. This makes it particularly useful when inverting large matrices is impractical or unnecessary for the entire system, and it allows for a deeper understanding of the contributions of specific subsystems to the overall dynamic behavior.

For a system subjected to harmonic excitation, where both the external force and the response oscillate at the same frequency $\omega$, the equations of motion can be simplified and analyzed in the frequency domain. The FRF matrix element $Hij$ $(\omega)$, representing the receptance between the i-th displacement (output) and the j-th force (input), can be calculated as follows:

$$H_{ij} = \frac{det(\Delta_{ij})}{det(\Delta)} (-1)^{i+j} \tag{32}$$

where $\Delta$ is the full dynamic stiffness matrix:

$$\Delta = -\omega^2[M] + i\omega[C] + [K] \tag{33}$$

and $\Delta_{ij}$ is the matrix obtained by removing the i-th column and j-th row from $\Delta$ while keeping the remaining elements unchanged. To derive the matrix $\Delta$, we consider the homo-

geneous algebraic equations obtained by taking the Laplace transform of the equations of motion. For the vehicle system, these equations are the following:

$$\begin{cases} \left(-\alpha_1\omega^2 + C_{sf}\omega j + K_{sf} + \frac{k_\theta}{L^2}\right)U_1 - \left(\alpha_2\omega^2 + \frac{k_\theta}{L^2}\right)U_2 - \left(K_{sf} + C_{sf}\omega j\right)U_3 = F_1 \\ -\left(\alpha_2\omega^2 + \frac{k_\theta}{L^2}\right)U_1 + \left(-\alpha_1\omega^2 + C_{sr}\omega j + K_{sr} + \frac{k_\theta}{L^2}\right)U_2 - (K_{sr} + C_{sr}\omega j)U_4 = F_2 \\ \left(-m_{uf}\omega^2 + \left(K_{sf} + K_{tf}\right) + \left(C_{sf} + C_{tf}\right)\omega j\right)U_3 - \left(K_{sf} + C_{sf}\omega j\right)U_1 = F_3 \\ \left(-m_{ur}\omega^2 + (K_{sr} + K_{tr}) + (C_{sr} + C_{tr})\omega j\right)U_4 - (K_{sr} + C_{sr}\omega j)U_2 = F_4 \end{cases} \quad (34)$$

From these equations, the coefficient matrix $\Delta$ is constructed as follows:

$$[\Delta] = \begin{bmatrix} \Delta_{11} & \Delta_{12} & \Delta_{13} & \Delta_{14} \\ \Delta_{21} & \Delta_{22} & \Delta_{23} & \Delta_{24} \\ \Delta_{31} & \Delta_{32} & \Delta_{33} & \Delta_{34} \\ \Delta_{41} & \Delta_{42} & \Delta_{43} & \Delta_{44} \end{bmatrix} \quad (35)$$

where

$$\begin{aligned} \Delta_{11} &= \left(-\alpha_1\omega^2 + C_{sf}\omega j + K_{sf} + \frac{k_\theta}{L^2}\right) \\ \Delta_{12} &= \Delta_{21} = -(\alpha_2\omega^2 + \frac{k_\theta}{L^2}) \\ \Delta_{13} &= \Delta_{31} = -\left(K_{sf} + C_{sf}\omega j\right), \\ \Delta_{22} &= \left(-\alpha_1\omega^2 + C_{sr}\omega j + K_{sr} + \frac{k_\theta}{L^2}\right), \\ \Delta_{33} &= -m_{uf}\omega^2 + \left(K_{sf} + K_{tf}\right) + \left(C_{sf} + C_{tf}\right)\omega j, \\ \Delta_{24} &= \Delta_{42} = (K_{sr} + C_{sr}\omega j), \\ \Delta_{44} &= -m_{ur}\omega^2 + (K_{sr} + K_{tr}) + (C_{sr} + C_{tr})\omega j \\ \Delta_{14} &= \Delta_{41} = \Delta_{23} = \Delta_{32} = \Delta_{43} = \Delta_{43} = 0 \end{aligned} \quad (36)$$

Thus, the determinant-based method simplifies the problem of finding the FRFs to computing the determinants of the dynamic stiffness matrix and its modifications for specific degrees of freedom. This classical technique is often preferred when a more direct inversion method is impractical or when FRFs are needed for specific degrees of freedom.

### 2.5.3. Modal Analysis and Subsystem Response Prediction

Modal analysis is a powerful framework for predicting the dynamic behavior of structures by decomposing complex systems into their fundamental modes of vibration. This approach is particularly useful for analyzing the receptance matrix, which can be expressed using modal superposition. The receptance matrix $H_{ij}(\omega)$, representing the relationship between the i-th displacement (degree of freedom) and the j-th force, is calculated as [28]:

$$H_{ij}(\omega) = \frac{X_i}{F_j} = \sum_{r=1}^{n} \frac{\phi_{ir}\phi_{jr}}{(\omega_r^2 - \omega^2) + 2j\xi_r\omega_r\omega} \quad (37)$$

where $\phi_{ir}$ and $\phi_{jr}$ are the mass-normalized mode shapes corresponding to the i-th and j-th degrees of freedom, respectively. $\omega_r$ and $\xi_r$ are the natural frequency and the damping ratio of the r-th mode.

The double-subscript notation (e.g., $\phi_{ir}$ and $\phi_{jr}$) refers to the mode shapes at specific degrees of freedom. The subscript i represents the degree of freedom where the response is measured, and j represents the degree of freedom where the force is applied.

To compute the mode shapes and corresponding parameters, the following equations are used:

$$\phi_r = \frac{1}{\sqrt{m_r}}\psi_r, \xi_r = \frac{c_r}{2\sqrt{k_r m_r}} \quad (38)$$

where $m_r$, $k_r$, and $c_r$ are the mass, stiffness, and damping coefficients associated with the r-th mode, respectively. These coefficients are calculated as follows:

$$m_r = [\psi_r]^T [M] [\psi_r], k_r = [\psi_r]^T [K] [\psi_r], c_r = [\psi_r]^T [C] [\psi_r] \tag{39}$$

While this method is efficient and effective for analyzing large systems, one significant limitation is mode truncation, where only a subset of modes is considered due to computational constraints or practical considerations. Truncating the modes can lead to inaccuracies in the predicted system response, particularly if the neglected modes have a significant impact on the system's dynamic behavior. Regarding damping, the modal approach presented here assumes general damping, though proportional damping is often employed in practical applications. Proportional damping assumes that the damping matrix [C] is a linear combination of the mass [M] and stiffness [K] matrices, which simplifies the analysis by ensuring that the mode shapes remain uncoupled. However, the half-car model analyzed in this study is approximately proportionally damped but not strictly so. There are small deviations from proportional damping, which could introduce minor discrepancies in the predicted dynamic response. Despite these deviations, the system behaves in a manner close to proportion, making the modal approach still applicable with a reasonable degree of accuracy.

## 3. Numerical Results and Validation

To evaluate the accuracy of the Frequency-Based Substructuring (FBS) method compared to classical techniques, a MATLAB (Version (2023b)) code was developed. This code calculates the receptance components, focusing on both magnitude and phase behaviors to provide a comprehensive assessment. Table 1 summarizes the input parameters used for the vehicle model.

**Table 1.** Input parameters used for modeling.

| Parameters | Value | Unit (SI) |
|:---:|:---:|:---:|
| CoG to FR wheel center, $L_f$ | 1.5 | m |
| Wheelbase, L | 3.2 | m |
| Sprung mass, $m_s$ | 900 | kg |
| Moment of inertia, $I_{CG}$ | 768 | $kgm^2$ |
| FR/RR unsprung mass, $m_{uf}/m_{ur}$ | 90/80 | kg |
| FR/RR tire damping, $C_{tf}/C_{tr}$ | 10/10 | Ns/m |
| FR/RR suspension stiffness, ksf/ksr | 23,000/18,000 | N/m |
| FR/RR tire stiffness, $k_{tf}/k_{tr}$ | 280,000/250,000 | N/m |
| FR/RR damping coefficient | 100/90 | Ns/m |
| Trimmed body angular stiffness, $k_\theta^A$ | 1000 | Nm/rad |
| Dummy mass, $m_d$ | 0.1 | kg |

Figures 4–13 present the comparison of predicted receptance (FRF) components obtained using the FBS method with those obtained using classical theoretical models. Both magnitude and phase responses are calculated across a frequency range that encompasses the system's natural frequencies. The magnitude response exhibits resonance peaks at natural frequencies and anti-resonance dips, while the phase response transitions smoothly across these frequencies. Resonance peaks indicate frequencies where the system exhibits significant vibrational response. Anti-resonances are points where the system's response is minimal due to destructive interference. Comparing the magnitude of receptance components using the FBS method with that using theoretical models highlights how well the FBS captures resonant frequencies. Near resonances, the phase typically transitions

through $-180°$, reflecting significant lag in the system's response relative to the input force. The phase plot provides insight into how the system's response evolves with frequency, indicating how the system behaves in terms of timing and damping effects. Near anti-resonances, the phase may shift back toward $0°$, showing a more in-phase relationship between the input and output.

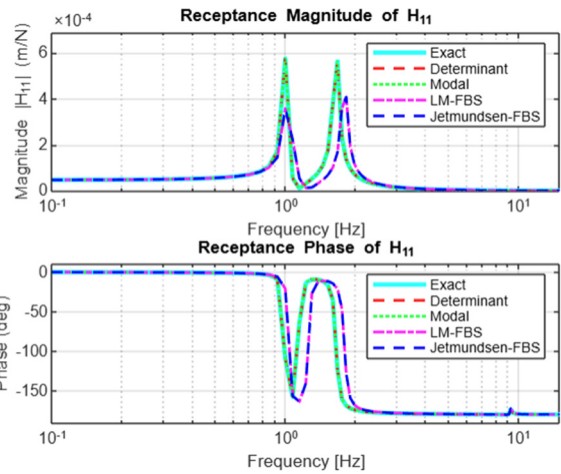

**Figure 4.** Magnitude and phase graph: receptance component of H11.

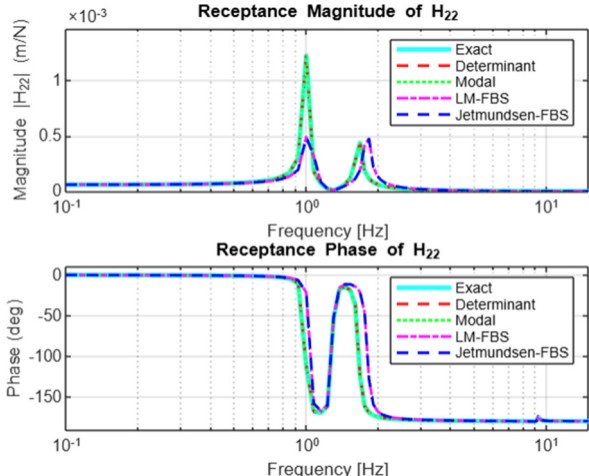

**Figure 5.** Magnitude and phase graph: receptance component of H22.

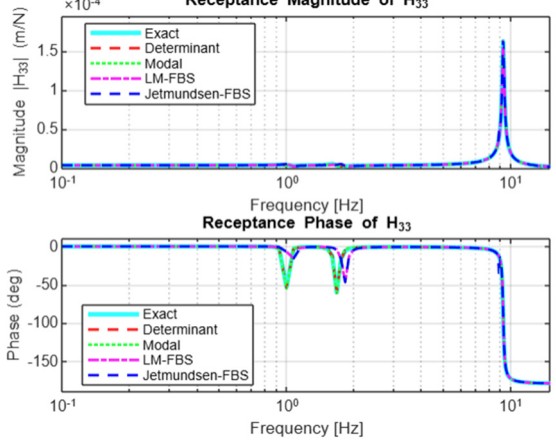

**Figure 6.** Magnitude and phase graph: receptance component of H33.

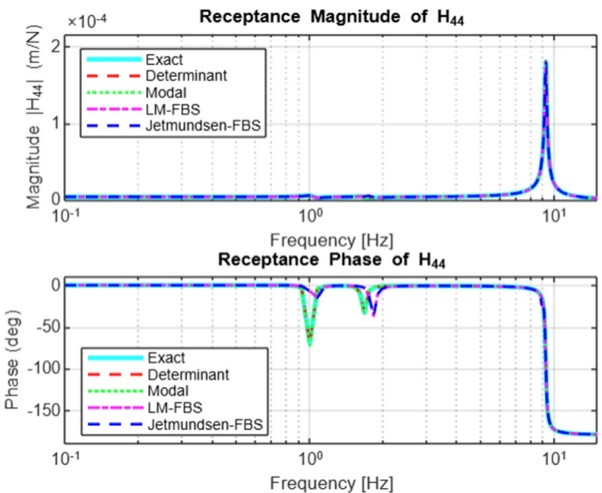

**Figure 7.** Magnitude and phase graph: receptance component of H44.

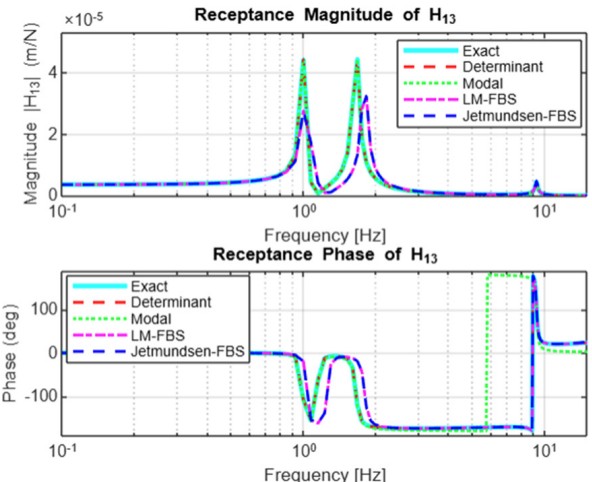

**Figure 8.** Magnitude and phase graph: receptance component of H13.

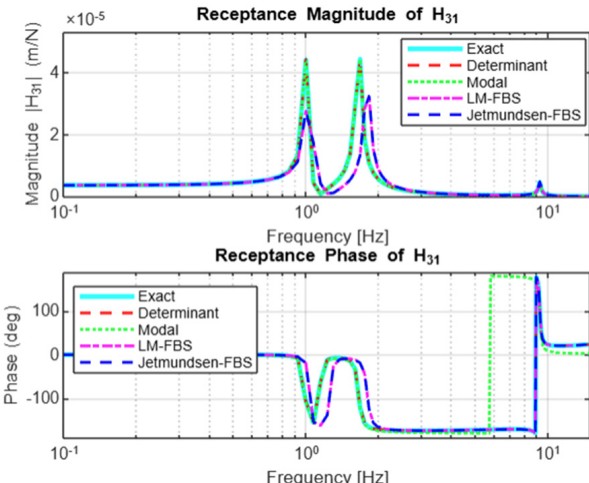

**Figure 9.** Magnitude and phase graph: receptance component of H31.

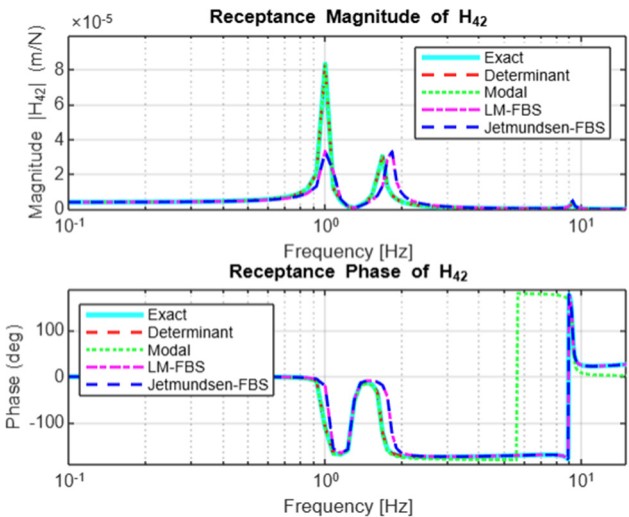

**Figure 10.** Magnitude and phase graph: receptance component of H42.

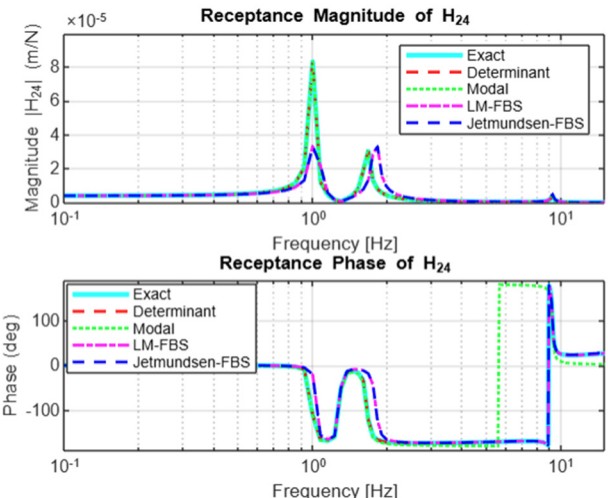

**Figure 11.** Magnitude and phase graph: receptance component of H24.

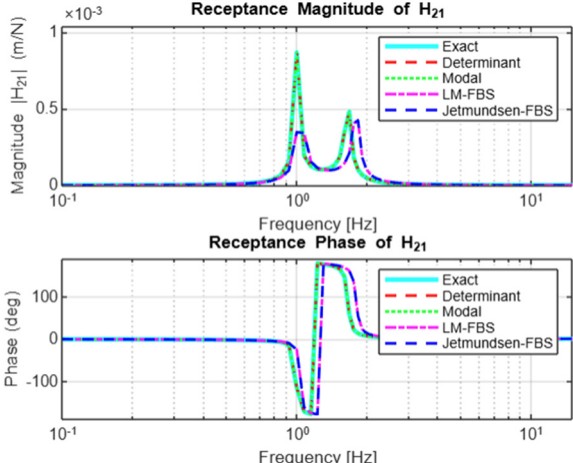

**Figure 12.** Magnitude and phase graph: receptance component of H21.

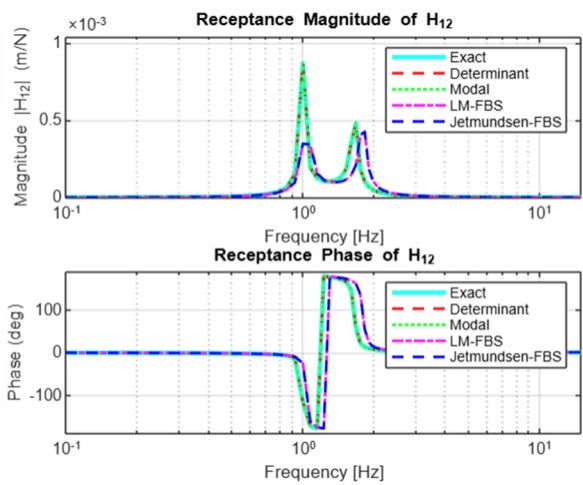

**Figure 13.** Magnitude and phase graph: receptance component of H12.

It is worth noting that the natural frequency ranges for pitch and bounce modes, as defined in the chassis handbook [29], provide a reference for validating the values in this analysis.

### 3.1. Receptance Component $H_{11}$ of the Coupled Car Model

The magnitude response of $H_{11}$, as shown in Figure 4, reveals distinct resonance peaks at approximately 1 Hz and 2 Hz, corresponding to the trimmed body bounce and pitch modes, respectively. Beyond these frequencies, the response diminishes significantly, indicating that higher-frequency modes contribute minimally to the system's dynamics. The dominance of the body dynamics is evident, as higher-frequency modes, such as unsprung mass dynamics, have negligible contributions to the system's overall response. The phase response transitions smoothly across the resonance frequencies, displaying a phase shift near these resonances. A notable drop toward $-180°$ occurs near the second resonance, which aligns with the expected dynamic behavior of coupled systems. Both the LM-FBS and Jetmundsen methods demonstrate strong agreement with the classical theoretical models, confirming their reliability in predicting dynamic behavior.

### 3.2. Direct Receptance Component $H_{22}$ of the Coupled Car Model

Similarly, the magnitude and phase analysis of $H_{22}$ follows a pattern consistent with $H_{11}$, with resonance peaks appearing around 1 Hz and 2 Hz. This component describes the rear interface response due to excitation applied at the front interface of the trimmed body and front suspension. The differences between $H_{11}$ and $H_{22}$, particularly in amplitude, can be attributed to variations in front and rear suspension properties and mass distribution. The phase plot for $H_{22}$ shows smooth transitions across the resonance frequencies, with a gradual decrease toward $-180°$. This behavior is consistent with dynamic systems subjected to damping and inertia effects. Both the LM-FBS and Jetmundsen methods align well with the exact and determinant-based methods, confirming their robustness for predicting coupled system dynamics (Figure 5).

### 3.3. Direct Receptance Component $H_{33}$ and $H_{44}$ of the Coupled Car Model

Figures 6 and 7 illustrate the magnitude and phase responses of $H_{33}$ and $H_{44}$, corresponding to the front and rear unsprung masses, respectively. The magnitude plots show clear resonance peaks around 9.2 Hz and 9.3 Hz, corresponding to the unsprung mass bounce modes, with minimal interaction from the trimmed body dynamics. This behavior is consistent across all methods, with both the LM-FBS and Jetmundsen-FBS methods providing accurate predictions of the system's dynamics. The phase response shows a sharp transition near these unsprung mass modes, dropping to $-180°$ and remaining constant beyond these frequencies. This reflects the dominant contribution of the

unsprung masses to the overall system behavior. The consistency of the results across all methods confirms the effectiveness of the FBS-based coupling techniques in capturing both magnitude and phase behavior accurately, particularly for systems dominated by unsprung mass dynamics.

*3.4. Cross-Receptance Component $H_{13}$, $H_{24}$, $H_{12}$ of the Coupled Car Model*

Figures 8 and 9 illustrate the cross-receptance components H31 and H13, which represent the interaction between the sprung and unsprung masses. These components exhibit good agreement across all methods. Resonance peaks appear around the natural frequencies of the trimmed body, while anti-resonance dips between these frequencies indicate the system's dynamic filtering effect. Both the LM-FBS and Jetmundsen algorithms align closely with the exact, determinant-based, and modal methods in capturing the magnitude behavior, particularly in both resonance and anti-resonance regions. The results show strong agreement between the LM-FBS and Jetmundsen algorithms and the exact and determinant-based methods, especially in the phase behavior for the first and second modes. These modes correspond to the bounce and pitch of the sprung mass, with natural frequencies of 1 Hz and 1.6 Hz, respectively. However, deviations are observed in the third and fourth modes, which relate to the bounce modes of the front and rear unsprung masses at 9.2 Hz and 9.3 Hz. The proximity of these two frequencies likely contributes to the phase discrepancies observed in the modal approach. As these unsprung mass bounce modes are very close in frequency, the modal method struggles to differentiate them, leading to slight phase inaccuracies. A similar pattern is observed for components $H_{24}$ and $H_{42}$ in Figures 10 and 11. For components $H_{21}$ and $H_{12}$, as shown in Figures 12 and 13, reciprocity holds, and both the magnitude and phase plots reflect typical dynamic behavior for a four-DoF system. Resonances occur around 1 Hz and 1.6 Hz, corresponding to the system's natural modes. The magnitude plots show distinct peaks at these frequencies, while the phase transitions smoothly across the resonances. All methods—including the exact, determinant, modal, LM-FBS, and Jetmundsen-FBS approaches—yield nearly identical results for these components, demonstrating the robustness of the receptance data. This consistency across methods confirms the accuracy of these coupling techniques in capturing the system's dynamic response, particularly in both resonant and anti-resonant behaviors. The results presented through the receptance analysis provide valuable insights into the dynamic behavior of the system, demonstrating a high degree of coherence across the methods used. The receptance components highlight the consistency and accuracy of the receptance coupling method using FBS, which are established based on the algorithms of LM-FBS and Jetmundsen when compared to classical theoretical approaches, validating their applicability in predicting system dynamics.

## 4. Conclusions

This study presents an effective application of the Frequency-Based Substructuring (FBS) method combined with the generalized receptance approach to predict the dynamic response of reconfigurable systems. The results from the receptance coupling methodology show strong agreement with theoretical models, demonstrating the robustness of the approach in capturing both resonant and anti-resonant behaviors of the system. Key findings indicate that the receptance coupling method reliably predicts both the magnitude and phase behavior across a wide frequency range, particularly at resonant and anti-resonant points. Resonance peaks in the magnitude response, observed at the system's natural frequencies, were accompanied by significant phase shifts approaching $-180°$, which is consistent with theoretical predictions for dynamic systems with low damping. The agreement between the predicted and theoretical natural frequencies at these resonance peaks further validates the method's accuracy. In contrast, the anti-resonance regions, where phase shifted back toward $0°$, demonstrated minimal displacement due to destructive interference, further corroborating the expected dynamic behavior.

For the specific vehicle parameters used in this simulation, the largest resonance peaks in magnitude were observed in the first and second modes, corresponding to the trimmed body bounce and pitch motions and reflected in the receptance component $H_{12}$. The smallest peaks were associated with the third and fourth modes, linked to the unsprung mass bounce motions. Validation of cross-receptance terms (e.g., $H_{12}$ and $H_{21}$) further supports the accuracy of the proposed method, confirming the system's linear and reciprocal behavior. The smooth phase transitions observed in these terms reinforce the method's consistency in predicting dynamic interactions between subsystems.

Numerically, the generalized receptance coupling approach matched closely with the results obtained from classical methods for the same system configuration. The agreement in natural frequencies and resonance peaks underscores the method's practical applicability, particularly for automotive and electric vehicle (EV) applications, where reliable dynamic predictions are essential in the early development phases.

Looking ahead, the receptance coupling method demonstrated in this study can be extended from a half-car model to more complex dynamic systems with flexible couplings and multiple substructures. Its capability to predict both resonant and anti-resonant behaviors, phase lags, and system response magnitudes positions this method as a valuable tool for addressing challenges in noise, vibration, and harshness (NVH) analysis. This is particularly relevant in the context of EV systems, where this methodology can be applied to a broader range of frequencies and more intricate structural configurations, including systems with higher damping or more complex boundary conditions.

**Author Contributions:** Conceptualization, B.H.; methodology, S.T.; software, B.H.; validation, B.H.; formal analysis, B.H.; investigation, B.H.; resources, S.T.; data curation, B.H.; writing—original draft preparation, B.H.; writing—review and editing, S.T.; visualization, B.H.; supervision, S.T.; project administration and funding acquisition for prototyping and testing. All authors have read and agreed to the published version of the manuscript.

**Funding:** This research received no external funding.

**Data Availability Statement:** The detailed data presented in this study are available on request from S.T.

**Conflicts of Interest:** The authors declare no conflicts of interest.

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
