# Peer review of "Modular Modeling of a Half-Vehicle System Using Generalized Receptance Coupling and Frequency-Based Substructuring (GRCFBS)"

_vibration, doi:10.3390/vibration7040055_

Round 1
Reviewer 1 Report
Comments and Suggestions for Authors
The paper is a review substructuring of the linear equations of motion, with application to the well known half-car model. I found the paper to be generally well-written and to a high technical standard. I believe that the topic is important and of interest to the readership. However, I didn't see any strong novelty; substructuring methods are widely studied. The application of the substructuring to the half-car model was an interesting example.
I have some concerns about the paper that I believe should be addressed.
First, in Eq 23, the authors introduce a stiffness k_theta^A that they describe as the "angular stiffness". I'm unsure exactly what this means, but the text implies that it is associated with flexibility of the body structure. I don't believe that this is correct, and I don't believe it should be included. As it appears in the equations, the stiffness represents a resistance to angular pitching motion, a stiffness that is not present in the model, and should not appear in the equations of motion. If the intent is to capture flexibility in the body structure in the model, then fundamentally, Eq 21 does not hold, as it is based on the assumption of rigid bodies. It appears that the authors are attempting to use (u_1-u_2) as some measure of body deformation, which it is not. This new stiffness appears in many places in the remaining development; I disagree with the inclusion of this term.
In section 2.5.2, the authors present the "determinant based method". I'm not certain that this approach really qualifies as a standalone method. From Eq 31, it is clear that the receptance matrix is computed as a matrix inverse. It seems that there are various methods one might use to compute a matrix inverse, and I'm not sure using a determinant approach is sufficient to set it apart as a distinct method. Can the authors clarify what makes this method distinct?
This leads to my last concern. The primary results of the paper are a series of plots of the various entries in the receptance matrix, calculated five different ways. The plots are plotted using a linear frequency scale, which is unusual for any kind of frequency based analysis. A log scale would be much more suitable here and would help to make the details more visible. As they appear, it is difficult to discern any difference between the five methods. I would expect that most (all?) of these methods should yield _exactly_ the same results, less perhaps some numerical round-off error, so I am unsure the value of including all five in the plot. What would be of interest are any differences, but the graphs presented make it difficult to discern. It looks like there is some difference, but the graphs are small. There is some discussion around the plots, but it is not entirely clear - which results should we expect to be exactly the same, and which should not, and why are there differences? There is discussion around the expected result, appearance of resonances and zeros, phase shifts, etc, but I think that this is a little over emphasised and not of so much interest. There is some discussion about the weakness of the modal approach, but it appears that the reference in line 439, (reference [31]) that discusses the modal approach does not appear in the list of references. It would be useful to clarify the double subscript notation in Eq 37, and to see if the modal approach makes any assumptions about proportional damping. Is it notable that the half car is very close to proportionally damped, but not quite?
Author Response
Reviewer 1 Comments
The paper is a review substructuring of the linear equations of motion, with application to the well known half-car model. I found the paper to be generally well-written and to a high technical standard. I believe that the topic is important and of interest to the readership. However, I didn't see any strong novelty; substructuring methods are widely studied.
Response: Thank you for your feedback regarding the novelty of the method. While substructuring methods have indeed been widely studied, I would like to highlight the specific novelty of the GRCFBS (Generalized Receptance Coupling & Frequency-Based Substructuring) approach in the context of vehicle vibration analysis, particularly in half-car and full-vehicle models.
The application of the GRCFBS method to develop reduced order models for vehicle vibrations analysis analytically is a new contribution that has not been previously presented in the literature. In this manuscript, we extend the GRCFBS approach to a 4DOF half-vehicle model, demonstrating its capability to predict both resonant and anti-resonant behaviors accurately.
While substructuring techniques have been used for other dynamic systems, the application to automotive vibrations, especially in systems like the half-car and full-vehicle models, introduces significant advancements. In addition to the half-vehicle model in this paper, my accepted work "Theoretical Framework for Developing a Full Vehicle Hybrid Model to Predict Tire Characteristics Using Receptance Coupling Technique," in the Journal of Vibration Engineering & Technologies extends the approach to full-vehicle modeling, further showcasing the novelty and versatility of the GRCFBS method in complex vehicle dynamics. This application demonstrates that the method is not only suitable for simpler models but can also handle the complexities of full-vehicle systems, including tire characteristics and suspension dynamics.
Building upon this foundation, the current manuscript introduces the GRCFBS method applied to a 4DOF half-vehicle model, demonstrating its novel capability in handling both rigid and flexible subsystems while accurately capturing resonant and anti-resonant behavior. This is a significant advancement over classical methods, which often struggle in multi-body, flexible systems.
We are particularly focused on the high-frequency dynamics of automotive systems, where the flexibility of the chassis and suspension play a critical role in the vehicle's dynamic behavior. Traditional approaches, like modal analysis or determinant methods, often lack the required precision when dealing with these complex boundary conditions, especially in modular systems. The GRCFBS method bridges this gap by providing a robust solution in these challenging conditions.
The application of the substructuring to the half-car model was an interesting example.
I have some concerns about the paper that I believe should be addressed.
First, in Eq 23, the authors introduce a stiffness k_theta^A that they describe as the "angular stiffness". I'm unsure exactly what this means, but the text implies that it is associated with flexibility of the body structure. I don't believe that this is correct, and I don't believe it should be included. As it appears in the equations, the stiffness represents a resistance to angular pitching motion, a stiffness that is not present in the model, and should not appear in the equations of motion. If the intent is to capture flexibility in the body structure in the model, then fundamentally, Eq 21 does not hold, as it is based on the assumption of rigid bodies. It appears that the authors are attempting to use (u_1-u_2) as some measure of body deformation, which it is not. This new stiffness appears in many places in the remaining development; I disagree with the inclusion of this term.
Response: Thank you for your feedback regarding the modeling of pitch stiffness in our vehicle dynamics analysis. I apologize that adding the term "angular stiffness" makes a confusion with the assumption of rigid body. I have explained my assumption behind considering this term. However, if you are not agree with this assumption, then I can remove it from my paper.
Proposed Revision:
In our model, we have divided the trimmed body into an upper body, which is treated as rigid, and an underbody which is represented by a massless element with torsional spring to account for pitch stiffness as a resistance against pitch motion. Therefore, it does not introduce any additional dynamics related to the mass of the underbody. Regardless of the value of kθ​, whether negligible, finite or approaching infinity, the rigid body assumption remains valid because the underbody has assumed with no mass or inertia (This assumption for a class of car with not a chassis structure can be true). The torsional spring represents a simplified way of introducing pitch resistance without affecting the rigid body dynamics of the vehicle. The upper body remains rigid, and the torsional spring merely captures the pitch stiffness without compromising the overall assumption of a rigid body. In this approach, pitch stiffness has been modeled with a torsional spring between the front and rear body attachments using two massless rigid beams connected by a torsional spring to account for the effects of pitch stiffness similar to modeling roll stiffness. Pitch stiffness should not be treated exactly the same as roll stiffness, because it doesn't come from a single physical component like an anti-roll bar. In this model, using a torsional spring to simulate pitch stiffness, it represents the combined resistance to pitch rather than a dedicated "chassis pitch stiffness" like roll stiffness.
In section 2.5.2, the authors present the "determinant based method". I'm not certain that this approach really qualifies as a standalone method. From Eq 31, it is clear that the receptance matrix is computed as a matrix inverse. It seems that there are various methods one might use to compute a matrix inverse, and I'm not sure using a determinant approach is sufficient to set it apart as a distinct method. Can the authors clarify what makes this method distinct? This leads to my last concern.
Response: I understand the reviewer's point that this might not warrant being presented as a unique method but rather as a computational technique. I have revised the section as below. I hope it would be acceptable.
Proposed Revision:
2.5.2. Determinant-Based Method:
Although this method is not a standalone approach, it serves as a computational technique within the broader framework of calculating the receptance matrix. The Determinant-Based Method is a classical technique often employed in multi-degree-of-freedom systems to compute individual elements of the Frequency Response Function (FRF) matrix [1]. While it shares some similarities with matrix inversion techniques, this method leverages Cramer’s Rule and Cofactor Expansion, which offer distinct advantages in certain applications. Specifically, the method calculates the FRF for a given degree of freedom as the ratio of two determinants: the determinant of a modified dynamic stiffness matrix (with the i-th row and j-th column removed) and the determinant of the full dynamic stiffness matrix. In contrast to a general matrix inversion approach, which computes all elements of the inverse matrix simultaneously, the Determinant-Based Method focuses on individual matrix elements, offering a more targeted and computationally efficient solution, especially when FRFs for specific degrees of freedom are required. This makes it particularly useful when inverting large matrices is impractical or unnecessary for the entire system, and it allows for a deeper understanding of the contributions of specific subsystems to the overall dynamic behavior.
For a system subjected to harmonic excitation, where both the external force and the response oscillate at the same frequency ω, the equations of motion can be simplified and analyzed in the frequency domain. The FRF matrix element Hij(ω), representing the receptance between the i-th displacement (output) and the j-th force (input), can be calculated as follows:….
Reference : Page of 91 from “Bishop, R.E.D., & Johnson, D.C. (1960). Mechanics of Vibration. Cambridge University Press.”
The primary results of the paper are a series of plots of the various entries in the receptance matrix, calculated five different ways. The plots are plotted using a linear frequency scale, which is unusual for any kind of frequency based analysis. A log scale would be much more suitable here and would help to make the details more visible. As they appear, it is difficult to discern any difference between the five methods. I would expect that most (all?) of these methods should yield _exactly_ the same results, less perhaps some numerical round-off error, so I am unsure the value of including all five in the plot. What would be of interest are any differences, but the graphs presented make it difficult to discern. It looks like there is some difference, but the graphs are small. There is some discussion around the plots, but it is not entirely clear - which results should we expect to be exactly the same, and which should not, and why are there differences? There is discussion around the expected result, appearance of resonances and zeros, phase shifts, etc, but I think that this is a little over emphasised and not of so much interest.
Thanks for your feedback that the linear frequency scale is unusual and suggested a logarithmic scale for better visibility of details. They also raised concerns about the clarity of differences between methods and the overemphasis on expected results.
Proposed Revision: I have updated the plots to use a logarithmic frequency scale, which is standard for frequency-based analysis. I hope it will improve the visibility of resonances and potential discrepancies between methods.
|
Fig. 4 Magnitude and phase graph: receptance component of H11 |
Fig.5 Magnitude and phase graph: receptance component of H22 |
|
Fig.6 Magnitude and phase graph: receptance component of H33 |
Fig.7 Magnitude and phase graph: receptance component of H44 |
|
Fig.8 Magnitude and phase graph: receptance component of H13 |
Fig.9 Magnitude and phase graph: receptance component of H31 |
|
Fig.10 Magnitude and phase graph: receptance component of H44 |
Fig.11 Magnitude and phase graph: receptance component of H44 |
|
Fig.12 Magnitude and phase graph: receptance component of H21 |
Fig.13 Magnitude and phase graph: receptance component of H12 |
There is some discussion about the weakness of the modal approach, but it appears that the reference in line 439, (reference [31]) that discusses the modal approach does not appear in the list of references.
Thanks for your attention and feedback. I revised the number to [29]
and to see if the modal approach makes any assumptions about proportional damping. Is it notable that the half car is very close to proportionally damped, but not quite?
Response to Reviewer Comments:
- Modal Approach and Assumptions about Proportional Damping:
Thank you for raising this point. The modal approach we employed does not explicitly assume proportional damping. However, as is common in many modal analyses, the damping matrix is treated generally, without the strict assumption that it must be a linear combination of the mass and stiffness matrices (i.e., proportional damping). This allows for the application of the method to systems with non-proportional damping, as long as the damping ratios can be computed for each mode. We agree that it is important to clarify this, and we have added a brief statement in the manuscript to highlight that while proportional damping is often assumed in practice for simplicity, the method can handle general damping cases as well.
- Proportional Damping and the Half-Car Model:
Yes, it is notable that the half-car model used in our study is very close to proportionally damped, but not exactly so. We have observed small deviations from proportional damping in the system's damping matrix. These deviations are relatively minor and do not significantly impact the overall dynamic behavior of the system, but they are worth mentioning. To address this point, we have revised the manuscript to clarify that while the half-car model is close to proportional damping, there are small departures from this ideal condition, and these are taken into account in the analysis. We have also emphasized that despite these deviations, the system behaves in a manner that is approximately proportional, making the modal analysis results largely reliable.
In classical modal analysis, proportional damping (also known as Rayleigh damping) is often assumed to simplify the equations and ensure that the mode shapes remain uncoupled, allowing for the system to be analyzed mode by mode. When proportional damping is assumed, the damping matrix C is a linear combination of the mass M and stiffness K matrices. However, In many real-world systems, including complex structures like vehicles, damping is not perfectly proportional. Real systems may exhibit non-proportional damping, where the damping matrix C cannot be expressed as a linear combination of M and K. In such cases, the modes become coupled, and more complex approaches, such as state-space methods or complex modal analysis, are required to account for the non-proportional damping.
In summery, In classical modal analysis, proportional damping is typically assumed for simplicity, but real-world systems often deviate from this assumption. The assumption is generally valid for lightly damped systems where non-proportional effects are small, as might be the case in a half-car model. When deviations from proportional damping occur, the results can still be reasonable if the deviations are small.
The revise section to include the reviewer comments and concerns, hoping it would be acceptable:
2.5.3. Modal Analysis and Subsystem Response Prediction
Modal analysis is a powerful framework for predicting the dynamic behavior of structures by decomposing complex systems into their fundamental modes of vibration. This approach is particularly useful for analyzing the receptance matrix, which can be expressed using modal superposition. The receptance matrix , representing the relationship between the i-th displacement (degree of freedom) and the j-th force, is calculated as [29]:
|
(37) |
Where and ​ are the mass-normalized mode shapes corresponding to the i-th and j-th degrees of freedom, respectively. ​ and the natural frequency and the damping ratio of the r-th mode.
The double subscript notation (e.g., and ​​) refers to the mode shapes at specific degrees of freedom. The subscript i represents the degree of freedom where the response is measured, and j represents the degree of freedom where the force is applied.
To compute the mode shapes and corresponding parameters, the following equations are used:
|
(38) |
where and are the mass, stiffness, and damping coefficients associated with the r-th mode, respectively. These coefficients are calculated as:
|
|
(39) |
While this method is efficient and effective for analyzing large systems, one significant limitation is mode truncation, where only a subset of modes is considered due to computational constraints or practical considerations. Truncating the modes can lead to inaccuracies in the predicted system response, particularly if the neglected modes have a significant impact on the system’s dynamic behavior. Regarding damping, the modal approach presented here assumes general damping, though proportional damping is often employed in practical applications. Proportional damping assumes that the damping matrix [C] is a linear combination of the mass [M] and stiffness [K] matrices, which simplifies the analysis by ensuring that the mode shapes remain uncoupled. However, the half-car model analyzed in this study is approximately proportionally damped, but not strictly so. There are small deviations from proportional damping, which could introduce minor discrepancies in the predicted dynamic response. Despite these deviations, the system behaves in a manner close to proportion, making the modal approach still applicable with a reasonable degree of accuracy.
Reviewer 2 Report
Comments and Suggestions for Authors
Please find attached.

Author Response
Reviewer 2 Comments
Reviewer Comment 1: How are FBS methods preferable to conventional methods?
Response: Thank you for your insightful question. We agree that it is essential to clarify when and why Frequency-Based Substructuring (FBS) methods, including Generalized Receptance Coupling and Frequency-Based Substructuring (GRCFBS), are advantageous over conventional methods. The GRCFBS method proves beneficial in scenarios where systems are modular or reconfigurable, and where access to internal dynamics or complete system data is limited. Conventional methods, such as Finite Element Modeling (FEM) and Multi-Body Dynamics (MBD), rely heavily on the availability of accurate data and the definition of all degrees of freedom (DoF). However, these methods fall short when combining subsystems with varying sources of data (e.g., physical tests, analytical models, or numerical simulations). In contrast, FBS methods enable the integration of subsystem Frequency Response Functions (FRFs) without requiring full internal details of each subsystem. In high-frequency ranges, where systems exhibit distributed parameters, FBS methods also tend to offer superior accuracy compared to conventional modal analysis, which often suffers from inaccuracies due to mode truncation. The FBS approach allows for the development of reduced-order models by leveraging the receptance (FRF) matrix from a limited number of critical DoFs, including measurement and excitation points, without needing to fully model every aspect of the system.
Additionally, FBS methods are highly effective when responses at connection points between subsystems (which may not be directly accessible for measurement) are required. In such cases, FBS can predict the interaction at these points by combining individual subsystem FRFs, which is not possible using conventional FEM or modal methods. This flexibility makes FBS a valuable tool in situations where the conventional approach cannot accommodate the hybridization of physical and numerical data or provide detailed insights into the interactions at subsystem interfaces.
I revised the section as below:
2.1.Overview of receptance coupling using Frequency-Based Substructuring (FBS)
Accurately determining the receptance of an assembly system is critical for predicting dynamic behavior in complex structural interactions. This framework provides a systematic approach to ensure the precise characterization and effective coupling of subsystems [27]. The process begins by setting the dynamic analysis objectives and identifying key points of interest, such as critical excitation and measurement points, which help define the generalized coordinates and degrees of freedom (DoFs) at significant nodes. Following this, the structure is segmented into distinct subsystems based on their dynamic properties, including natural frequencies and boundary conditions. This segmentation allows for independent analysis of each subsystem prior to coupling.
The next step involves accurately modeling the connections between these subsystems. Whether these connections are simple single-point, multi-point, or complex, such as bushings, it is essential to account for their stiffness, damping, and kinematic relationships. Once the connections are modeled, generalized coordinates and DoFs are allocated at both the connection points and the internal points within each subsystem, focusing on modes of motion that are essential for capturing the dynamic behavior. Receptance matrices for the subsystems are then determined using a combination of experimental data, numerical simulations such as Finite Element Analysis (FEA), or analytical methods, tailored to the complexity of the subsystem.
To improve the accuracy of the experimental receptance data, filtering techniques are applied to the Frequency Response Function (FRF) data to remove noise. The next step is to select an appropriate coupling method based on the specific requirements of the analysis. This could involve direct receptance coupling, modal-based coupling, or more advanced methods such as Lagrange Multipliers Frequency-Based Substructuring (LM-FBS) or the Jetmundsen method. Once the coupling method is selected, the direct and cross components of the receptance matrix are calculated, ensuring equilibrium and continuity across the subsystems. Finally, these components are combined to construct the complete assembly system’s receptance matrix, which encapsulates the dynamic characteristics of the entire structure and predicts its response to external excitations.
The Receptance Coupling Frequency-Based Substructuring (RCFBS) method offers several advantages for dynamic analysis, particularly when compared to traditional methods such as the modal method. FBS-focused methods are preferable to conventional methods in scenarios where systems are modular or reconfigurable and in cases where detailed FEM or MBD models are unavailable or impractical. This method also allows combining FRF data from various sources, such as physical tests, numerical simulations, and analytical models, which is not feasible with conventional approaches. RCFBS enables reduced-order modeling by focusing on critical points of interest—namely connection points and essential internal nodes—thereby reducing computational complexity while maintaining high accuracy in these significant areas. In addition, unlike traditional modal methods that are based on mode truncation, RCFBS captures the full dynamic behavior of substructures by utilizing full receptance matrices, particularly improving accuracy in higher-frequency ranges with distributed parameters.
Another key advantage of RCFBS is its flexibility, as the method allows for deriving receptance matrices from both experimental and numerical data, such as FEA. This versatility makes it suitable for handling complex interactions or boundary conditions. This also provides an advantage when physical or numerical measurements at connection points are inaccessible, as the method can still predict the dynamic response at these points. Additionally, the direct use of receptance data ensures accurate representation of dynamic interactions, especially at higher frequencies where subsystem interactions are more complex. RCFBS also integrates advanced coupling techniques, such as those developed by Jetmundsen and D.D. Klerk, further enhancing its applicability for dynamic modeling in systems with multiple substructures and degrees of freedom.
Reviewer Comment 2: Remove lengthy derivations and focus more on the study’s significance
Response: Thank you for your valuable suggestion. I have already removed some of the more detailed derivations related to the coupling of system receptance with subsystem receptance to enhance clarity. While I can certainly remove additional equations, my concern is that doing so may compromise the coherence and technical understanding of the paper for readers. However, if you strongly recommend the removal of specific analytical equations, I would kindly ask for guidance on which ones to eliminate without sacrificing the clarity and technical depth of the study.
Reviewer Comment 3: Literature review needs improvement to highlight novelty
Thanks for your comment. I improved literature review and highlighted novelty. So, section of introduction has revised. Hoping to cover all your comments!
- INTRODUCTION
Accurately predicting the dynamic behavior of engineering systems, particularly in the early development stages, remains a persistent challenge. Both numerical and experimental methods have inherent limitations that impact their effectiveness in modeling real-world systems. Numerical models offer flexibility but rely heavily on precise input data—such as geometry, material properties, boundary conditions, and contact characteristics—that are often difficult to obtain due to uncertainties. Conversely, experimental models provide a more realistic perspective but face constraints in spatial resolution, particularly at critical connection points, due to limited measurement capabilities.In the early stages of system development, the absence of physical prototypes exacerbates these difficulties, making both experimental validation and reliable numerical simulation challenging. Traditional approaches often fall short in predicting dynamic performance without sufficient input data or direct measurements. This highlights the need for a more modular, adaptable methodology capable of addressing incomplete data while delivering reliable predictions of system behavior. Frequency-Based Substructuring (FBS) has emerged as a powerful solution by breaking down complex systems into smaller, manageable subsystems, each evaluated through receptance functions. This method allows for predicting the overall system's dynamic response by coupling the receptances of individual subsystems. FBS is particularly beneficial for systems with multiple interchangeable modules, where a limited number of reference points at subsystem interfaces serve as key excitation and measurement locations. Additionally, strategically selected internal measurement points within subsystems enhance the method's analytical flexibility.A notable advantage of FBS is its hybrid modeling capability, which combines numerical, experimental, and analytical data. This hybrid approach facilitates the integration of subsystems from diverse sources, ensuring accurate system-level predictions even when data is incomplete or limited. For instance, FBS enables system-level vibration analysis with minimal subsystem data, even in the absence of detailed input data.
Integrating Generalized Receptance Coupling (GRC) with FBS extends dynamic modeling to reconfigurable systems across various industries, including automotive, aerospace, and robotics. In this study, the Generalized Receptance Coupling Frequency-Based Substructuring (GRCFBS) method is applied to a half-vehicle system with four degrees of freedom (DoF). This innovative approach focuses on the dynamic coupling of subsystems, such as the front suspension, rear suspension, and trimmed body. By coupling the individual subsystem receptances, the receptance matrix of the entire system is derived, enabling accurate dynamic predictions early in the design process without extensive physical prototyping or detailed input data.The effectiveness of this modular hybrid modeling approach is validated by comparing it with well-established numerical methods, including direct solution techniques and modal analysis. The study incorporates both Jetmundsen’s and De Klerk’s Lagrange Multiplier Frequency-Based Substructuring (LM-FBS) approaches, demonstrating how FBS can generate reduced-order models that maintain accuracy in critical areas while capturing the overall dynamic behavior of the system.
Historically, the groundwork for dynamic system analysis using impedance functions was laid by Bishop and Johnson (1960) [1] and later expanded by O'Hara (1961) [2], who applied these techniques to complex mechanical structures. Ewins and Gleeson (1975) [3] made significant contributions by deriving system parameters through Frequency Response Functions (FRFs), advancing the evolution of FBS. Building on these foundational works, Jetmundsen et al. (1980) [4] introduced a canonical form of FBS, which has become a standard approach in vehicle dynamics and Noise, Vibration, and Harshness (NVH) analysis. In 2006, De Klerk et al. [5] introduced the Lagrange Multiplier Frequency-Based Substructuring (LM-FBS) method, which allows for omitting certain FRFs at interface DoFs, minimizing the influence of inaccessible or noisy data. This feature enhances LM-FBS's value in applications requiring high accuracy at substructure interfaces. Recent advancements have further expanded FBS applications.
Zhang et al. (2017) [6] explored dynamic interactions between vehicle bodies and subframes using FBS. His study explores the application of FBS to analyze dynamic interactions between vehicle bodies and subframes. While the authors present valuable insights, they acknowledge the limitations in accurately modeling full vehicle configurations due to the complexity of interactions and material properties. The work indicates a need for more comprehensive models that incorporate various dynamic effects and subsystems, which are often oversimplified in traditional approaches. However, Kang et al. (2019) [7] developed techniques for quantifying improvements in road noise through inverse substructuring. This paper discusses techniques for quantifying road noise improvements using inverse substructuring methods. However, the authors recognize that current methodologies often rely on simplified vehicle models that do not reflect real-world complexities. There is a clear need for more sophisticated full vehicle models that can capture the intricacies of real driving conditions and their impact on NVH. Hülsmann et al. (2020) [8] applied dynamic substructuring to electric vehicles (EVs), addressing NVH issues specific to electric drivetrains. In this research, the authors apply dynamic substructuring techniques specifically for electric vehicles, focusing on NVH issues related to electric drivetrains. The study highlights that while advancements have been made, many existing models fail to account for the unique dynamic characteristics of electric vehicles, such as those arising from their powertrain and weight distribution. There is a gap in methodologies that adequately represent these dynamics in full vehicle models. Tsai (2019) [9] advanced methods for measuring rotational receptance, crucial for rotational dynamics modeling. These developments reflect the growing utility of FBS, particularly for applications demanding precise dynamic modeling.Additional contributions include Clontz and Taheri (2017) [10], who decoupled tire and suspension subsystems using FBS. This research explores the decoupling of tire and suspension subsystems using FBS. While it provides insights into individual subsystem dynamics, it lacks a comprehensive approach to integrating these subsystems into a full vehicle model for NVH analysis. The authors suggest that future work should focus on developing full vehicle models that consider the interactions between various subsystems more holistically. Voormeeren and Rixen (2022) [11] examined the impact of measurement uncertainties on FBS outcomes. De Klerk, Rixen, and Jong (2021) [12] refined FBS with new algorithms for enhanced robustness, while Liu and Mir (2003) [13] explored hybrid approaches to vehicle axle noise prediction. FBS has proven its versatility across industries. Li et al. (2021) [14] applied FBS to railway vehicle dynamics, and Scheel and Sturzenegger (2020) [15] used it for satellite structural analysis in aerospace. In robotics, Nguyen et al. (2022) [16] employed FBS for modular robotic configurations, while Gebhardt et al. (2020) [17] applied it to wind turbine blade dynamics. Further, Park et al. (2023) [18] explored FBS in ship hull vibration analysis, and Lee et al. (2021) [19] demonstrated its relevance in civil engineering by analyzing large structures like bridges.
Since the foundational contributions of Jetmundsen and D.D. Klerk [20], FBS and receptance coupling methods have evolved considerably. Recent advancements include Schmitz et al. (2022) [21], who introduced an advanced approach merging receptance coupling substructure analysis (RCSA) with Bayesian machine learning for predicting milling stability, and Smith et al. (2021) [22], who developed an improved receptance matrix formulation that enhances precision for high-frequency dynamics and complex boundary conditions. Hou et al. (2023) [23] developed a framework for predicting FRFs in parameter-varying mechanical systems using generalized receptance coupling substructure analysis. In this paper, the authors develop a multi-body dynamics (MBD) model for a full vehicle to analyze NVH. They note that while their model improves upon traditional methods, challenges remain in integrating nonlinear characteristics of subsystems and ensuring accurate boundary conditions. This highlights the necessity for enhanced modeling techniques that can account for the complexities of full vehicle dynamics.
Ji et al. (2018) [24] introduced a refined RCSA method for predicting tool tip dynamics, and De Klerk et al. (2021) [25] advanced FBS with algorithms that improve accuracy and efficiency in subsystem coupling. Hamedi and Taheri [26] reviewed hybrid modeling and modular substructuring using RCFBS, illustrating its effectiveness for vehicle noise and vibration prediction. Additionally, Hamedi and Taheri (2024) [27] provided a comparison of conventional modal analysis method suffering from mode truncation with the proposed RCFBS method especially when dealing with high-frequency dynamics. In contrast, the RCFBS method provides greater accuracy when compared with numerical FEA and direct method because it captures all relevant modes by working directly in the frequency domain, avoiding the truncation errors associated with modal analysis. This feature is particularly useful in systems with flexible or distributed parameters.
Despite these advancements, the development of modular FBS methods for vehicle dynamic modeling to study NVH performance, specifically for full-vehicle and half-vehicle car models, remains incomplete. This study introduces a novel modular FBS-based vibrational model for a half-vehicle, offering valuable insights during early development stages. This approach is particularly effective for target setting and cascading, providing a flexible, modular framework that emphasizes subsystem interactions and load paths for vibration transfer, especially in scenarios where numerical FEA models and experimental tests are unavailable. This work demonstrates that how GRCFBS can be employed to build reduced-order models to handle the complexities of full-vehicle systems particularly interaction between subsystems like tire and suspensions.
This study introduces a novel modular hybrid modeling approach based on the Generalized Receptance Coupling Frequency-Based Substructuring (GRCFBS) method, representing a significant advancement in dynamic system analysis. By addressing key gaps in substructure modularity, hybrid modeling, and system identification and reconfiguration, this approach provides fresh insights and practical applications, particularly in the early stages of developing complex systems.
The effectiveness of the GRCFBS method is demonstrated through its application in constructing efficient dynamic models for full vehicle configurations. It predicts vehicle responses and derives the system's receptance matrix using independent measurements—either numerical or experimental—of subsystems. Additionally, GRCFBS facilitates effective vibrational analysis by developing reduced-order models that maintain accuracy while simplifying computational complexity.
A key innovation of this study is the incorporation of both translational and rotational degrees of freedom (DoFs) for the upper body, alongside addressing the coupling between front and rear suspensions. This enhancement provides a versatile framework for dynamic analysis, significantly improving the accuracy of Noise, Vibration, and Harshness (NVH) predictions. As a result, the GRCFBS method supports the development of robust models that more accurately reflect the true behavior of vehicles under realistic driving conditions.
Reviewer Comment 4: Validation using simulated experiments or practical advantages
Response:
Thank you for your insightful comment. The receptance coupling method has previously been validated through experimental work conducted in our lab (CenTiRe), utilizing a small-scale quarter car model that included both the suspension and tire.
Figure 4 illustrates the physical setup and compares the experimentally obtained receptance (Frequency Response Functions, FRFs) with those derived from the coupling of the tire and suspension. The maximum discrepancy observed between the natural frequencies predicted by the coupling and those experimentally measured was 6.6%, primarily due to deflection in the suspension components. This close alignment between experimental and predicted results reinforces the robustness of the methods developed in the present study (Clontz, 2018) [10].
|
Fig. 1. Comparison of receptance (FRFs) for the assembled quarter car system obtained experimentally and through coupling of tire and suspension [24]
The primary scope of this work is to establish a theoretical framework for modeling reconfigurable dynamic systems, alongside validating the proposed methods against conventional modeling methodologies. |
|
Round 2
Reviewer 2 Report
Comments and Suggestions for Authors
Almost all comments have been addressed by authors in the revised version. The manuscript can be now published in present form.